# Saffron extract attenuates neuroinflammation in rmTBI mouse model by suppressing NLRP3 inflammasome activation via SIRT1

**Mariam J. Shaheen[1], Amira M. Bekdash[2], Hana A. Itani[2,3]\*, Jamilah M. Borjac[1]\***

1 Department of Biological Sciences, Beirut Arab University, Debbieh, Lebanon, 2 Faculty of Medicine, Department of Pharmacology and Toxicology, American University of Beirut, Beirut, Lebanon, 3 Division of Clinical Pharmacology, Department of Medicine, Vanderbilt University Medical Center, Nashville, Tennessee, United States of America

\* hi40@aub.edu.lb (HAI); j.borjac@bau.edu.lb (JMB)

**Data Availability Statement:** All relevant data are within the manuscript and its Supporting Information files and authors are ready to provide details without limitations.

## Abstract

Traumatic brain injury (TBI) remains a major cause of morbidity and disability worldwide and a healthcare burden. TBI is an important risk factor for neurodegenerative diseases hallmarked by exacerbated neuroinflammation. Neuroinflammation in the cerebral cortex plays a critical role in secondary injury progression following TBI. The NOD-like receptors (NLR) family pyrin domain containing 3 (NLRP3) inflammasome is a key player in initiating the inflammatory response in various central nervous system disorders entailing TBI. This current study aims to investigate the role of NLRP3 in repetitive mild traumatic brain injury (rmTBI) and identify the potential neuroprotective effect of saffron extract in regulating the NLRP3 inflammasome. 24 hours following the final injury, rmTBI causes an upregulation in mRNA levels of NLRP3, caspase-1, the apoptosis-associated speck-like protein containing a CARD (ASC), nuclear factor kappa B (NF-κB), interleukin-1Beta (IL-1β), interleukin 18 (IL-18), nuclear factor erythroid 2–related factor 2 (NRF2) and heme oxygenase 1 (HMOX1). Protein levels of NLRP3, sirtuin 1 (SIRT1), glial fibrillary acidic protein (GFAP), ionized calcium-binding adaptor molecule 1 (Iba1), and neuronal nuclei (Neu N) also increased after rmTBI. Administration of saffron alleviated the degree of TBI, as evidenced by reducing the neuronal damage, astrocyte, and microglial activation. Pretreatment with saffron inhibited the activation of NLRP3, caspase-1, and ASC concurrent to reduced production of the inflammatory cytokines IL-1β and IL-18. Additionally, saffron extract enhanced SIRT1 expression, NRF2, and HMOX1 upregulation. These results suggest that NLRP3 inflammasome activation and the subsequent inflammatory response in the mice cortex are involved in the process of rmTBI. Saffron blocked the inflammatory response and relieved TBI by activating detoxifying genes and inhibiting NLRP3 activation. The effect of saffron on the NLRP3 inflammasome may be SIRT1 and NF-κB dependent in the rmTBI model. Thus, brain injury biomarkers will help in identifying a potential therapeutic target in treating TBI-induced neurodegenerative diseases.

**Funding:** This study has been supported by MS, the first author of this paper, who purchased the kits needed to measure the cytokines levels, the primers, and the kits needed to measure the gene expression, and by HAI through the AUB Medical Practice Plan fund who covered for the reagents used in western blot experiment along with 3 antibodies (NLRP3, SIRT1, and Beta tubulin). The lab assistant AB received a salary from the American Society of Nephrology (ASN), Carl W. Gottschalk Research Scholar Award (M0048845), and the American University of Beirut Faculty of Medicine Medical Practice Plan (MPP 320174). The funders had role in study design, data collection and analysis, decision to publish, or preparation of the manuscript.

**Competing interests:** The authors declare no competing interests.

## Introduction

Traumatic brain injury (TBI) is a complex of neurological complications introduced by diverse mechanical forces applied to the brain. Globally, each year, an estimation of 69 million people of different ages suffer from TBI and its related outcomes (death, physical and cognitive disabilities) [1]. Repetitive mild traumatic brain injury (rmTBI) or concussion is, according to the International Consensus Conference on Concussion in Sports "a complex pathophysiological process affecting the brain, induced by biomechanical forces" [2]. rmTBI is frequent among environmental workers, older people with imbalance disorders who frequently fall at home, athletes, or motor vehicle accidents [3]. TBI is described by primary and secondary injuries [4]. The primary injury is the mechanical displacement of tissue held at the point of trauma and no therapy interventions are applicable [5]. The secondary damage aggravates the immediate shock in a time-dependent manner extending to the entire head within days, weeks, or even years [6]. Though the exact processes behind TBI development continued to be uncertain, the secondary injury has been admitted to be the only key controlling TBI progression. Post-traumatic inflammation or neuroinflammation is one of the primary complications developed after TBI and is a prominent contributor to neurodegeneration development and neurological impairments [7]. It is marked by an increase in neuronal cell death, glial cell activation, recruitment of peripheral immune cells such as neutrophils and monocytes that can cross the blood-brain barrier, and excessive release of inflammatory mediators within the brain [7]. Cellular death or damage provokes the release of ions, molecules, and proteins described as the damage-associated molecular patterns (DAMPs) [8]. After TBI, DAMPs are recognized by the pattern recognition receptors (PRRs) [9] that activate the inflammasomes, a multi-protein complex structure, known as the Nucleotide-binding oligomerization domain, Leucine-rich Repeat and Pyrin domain-containing NLRP [10]. NLRP3 functions as a PRR, it mediates caspase 1 activation leading to the production of pro-inflammatory cytokines such as interleukin-1$\beta$ (IL-1$\beta$) and interleukin-18 (IL-18) and inducing pyroptosis [11]. NLRP3 participates in many types of neurological conditions accompanied by inflammation including TBI [12].

A growing body of research has shown that neuroinflammation is directly linked to the Sirtuin protein family specifically SIRT1 [13]. Sirtuins are NAD$^+$ dependent deacetylases [14]. Studies have shown SIRT1 is involved in many biological functions including inflammatory reaction, oxidative stress, mitochondrial biogenesis, and programmed cell death [15,16]. Recently, in TBI and associated neurodegenerative complications, Sirtuin showed to be a promising therapeutic target through the involvement of the NF-$\kappa$B and the NLRP3 inflammasome pathways [17].

Increased attention to NLRP3 as a new TBI therapeutic target is being addressed in new studies. The presence of NLRP3 in different models such as in Alzheimer's disease, stroke, NLRP3 knock-down or knock-out animal models was shown to enhance the neuroinflammatory response [18,19]. A growing number of studies have found that naturally occurring compounds such as mangiferin [20], omega-3 fatty acids [21], and apocynin [22] that target this inflammasome directly or indirectly, may minimize its activity following TBI. *Crocus sativus L.* popularly known as *Saffron or Z*aferan is a flower belonging to the iridaceous family that originated and cropped mainly in Iran besides Southeast Asia and North Europe [23], maybe a potential therapy for treating TBI and ameliorating its outcome. The phytochemical studies on saffron showed four major metabolites, crocin (80%), safranal (70%), and a modest percentage of picrocrocin and crocetin. The chemical analysis revealed over 200 distinct compounds found in saffron including flavonoids, amino acids, vitamins (riboflavin and thiamine), anthocyanin, proteins, α-carotene, β-carotene, starch, zeaxanthin, mineral matter, and gums [24].

The two major carotenoids crocin and crocetin are primarily metabolized in the intestinal tract [25]. In the intestine, crocin is enzymatically deglycosylated to trans-crocetin [26] enzymatically by the fecal microbiota [27], then crocetin is absorbed in the intestine [28] into the bloodstream, via passive transcellular diffusion, or in the liver at which it will be converted to glucuronide and crocetin glucuronide conjugates [27]. Crocetin, in the bloodstream, is bound to albumin and the albumin-bound crocetin pushes through the blood-brain barrier (BBB) slowly enough to enter the CNS [29]. The medicinal properties of saffron have been known for over 2500 years [30] and the pharmacological studies on saffron classified it as a safe and low toxic substance [31]. Saffron exhibits antioxidant properties [32] and anti-inflammatory activities [33] via its major metabolites crocin [34,35], crocetin [36], and safranal [37]. The neuroprotective role of saffron in neurodegenerative disorders such as Alzheimer's disease (AD) [38], Parkinson's disease (PD), multiple sclerosis, and TBI [39] has been confirmed by increased research in both laboratory models and patient clinical trials [40]. In the present study, we aim to investigate the role of NLRP3 in the rmTBI model and explore the possible neuroprotective effect of saffron extract in regulating NLRP3 inflammasome and SIRT 1 expression in rmTBI.

## Materials and methods

### 1. Preparation of saffron water extract

Saffron grade 1 stigma (Crocus sativus L.) was purchased from Novin Saffron Company provided by Mehran Trading Company (Mashhad Iran, ISIC code: 4721) that was cultivated and collected according to the WHO guidelines on good agricultural and collection practices (GACP) for medicinal plants [41]. A 2% aqueous solution was prepared by soaking one gram of ground saffron stigmas in 50 ml of distilled water for 3 hours at room temperature in the dark. The filtrate was then freeze-dried yielding 0.065 g of a yellow-orange precipitate. The precipitate was suspended in saline to a final concentration of 8.6% and used at a dose of 50mg/kg of body weight. The saffron dose used was based on animal studies showing its safety and effectiveness in neurologically related disorders and other disease [30,42–46].

### 2. Experimental groups and rmTBI induction

Male albino BABL/c mice weighing 30–40 g were obtained from the animal house of Beirut Arab University (Debbieh, Lebanon). Mice were housed under standard laboratory conditions of light (12-hour light/dark cycle), humidity, and temperature and had ad libitum access to standard mouse diet and tap water. The animal protocol was conducted and approved by the Institutional Review Board (IRB) at Beirut Arab University (code number 2019A-0039-S-P-0334) that complies with The Canadian Council on Animal Care's (CCAC) Guide to the Care and Use of Experiment animals [47]. The study was carried out in compliance with the ARRIVE guidelines. All of the induced injuries were performed after anesthetizing the mice with an intraperitoneal injection of a combination of xylazine (10 mg/kg, Interchemie Weken, Holland), and ketamine (50 mg/kg, Sterop Belgium) to minimize suffering or pain. A total of 36 mice were randomly divided into four groups (1) sham, (2) saffron sham, (3) TBI, (4) saffron TBI (n = 9 in each group). TBI was induced using a weight drop model setup in our laboratory modified from Feeney's method [48]. A 40g, 10mm diameter metal freely falling from an 18 cm height plexiglass tube hit the mouse closed head. The mice were placed on an iron platform without any head fixation, positioned to induce a diffuse injury (Fig 1). Mice were subjected to 7 hits over 14 days with 48-hour intervals. The saffron extract used in Saffron sham (Group 2) and Saffron TBI (Group 4) was administered intraperitoneally, 30 minutes

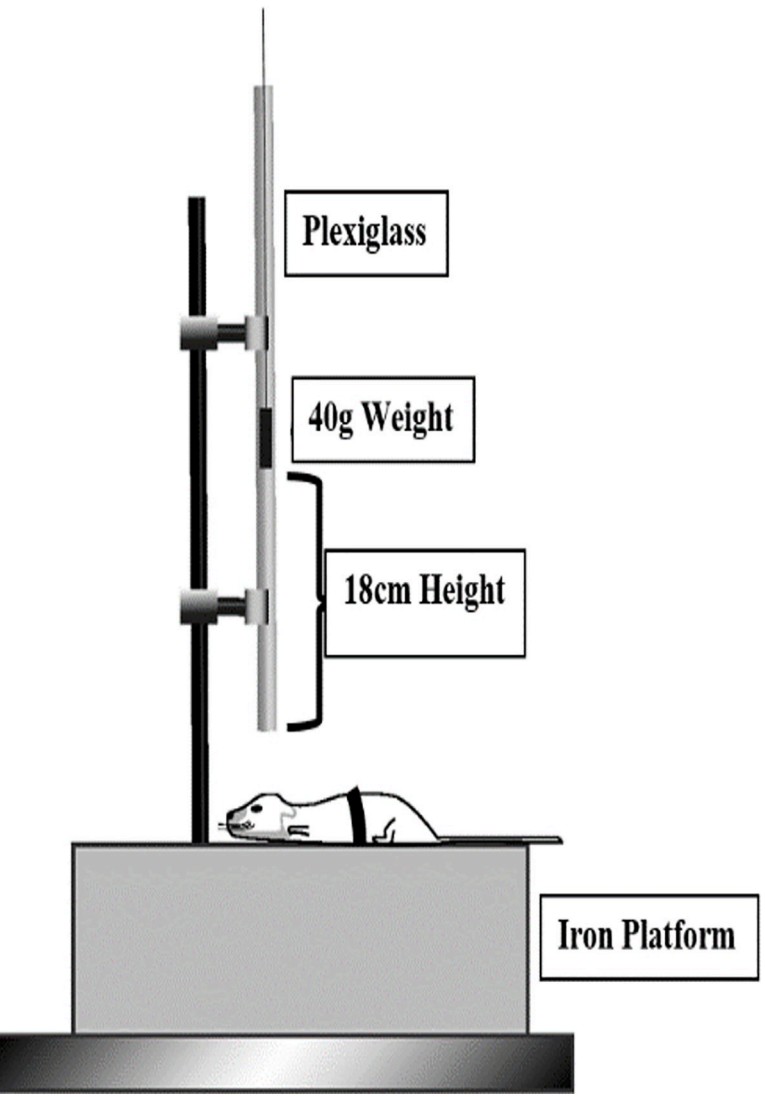

**Fig 1. Scheme of TBI apparatus used to induce brain trauma.** A 40 gram 10–mm diameter metal is fallen freely on a 50 cm plexiglass tube at 18 cm in height. Both, untreated and treated TBI mice were positioned on an iron platform without any head fixation, to induce a diffuse injury. The injury was repeated seven times over 14 days and 48 hours intervals between each injury.

before every hit. The body weight of each mouse was determined before saffron administration to ensure that mice receive the exact amount. After receiving the brain injury, mice were laid on heat pads for 10 min before being returned to their home cage. At the end of the study, before dissection, all the mice were anesthetized by an intraperitoneal injection of a combination of xylazine (10 mg/kg) and ketamine (50 mg/kg). Mice were dissected 24 hours post the final injury. The hearts were perfused with phosphate buffer saline, and the brain cortices were collected, immediately frozen in liquid nitrogen, and stored at -80˚C until use for molecular studies. Any mouse exhibiting fatigue and inflammation signs or any deformation that led to biased results was excluded from the study. To minimize potential confounders' effect, tasks were performed by the same researcher each time and all mice in their cages were returned to their same location.

## 3. Tissue processing

Brain cortices were mechanically homogenized in lysis buffer (PBS 0.01M, pH7.4 & 1mM PMSF) at a ratio of 1:9 (w/v) and then centrifuged for 5 min at 5000xg. The supernatants were subsequently used for enzyme-linked immunosorbent assay (ELISA), RNA extraction, protein quantification, and western blot analysis.

## 4. Real-time polymerase chain reaction (RT-PCR)

RNA was isolated from the brain cortices using the RNeasy Mini Kit (Qiagen, Germany). The RNA was then used for the reverse transcriptase reaction using QuantiTect Reverse Transcription Kit (Qiagen, Germany). Gene quantification was performed using qRT-PCR using the primers listed in Table 1 for NLRP3, ASC, caspase 1, interleukin-18 (IL-18), interleukin-1Beta (IL-1β), nuclear factor erythroid 2–related factor 2 (NRF2), nuclear factor kappa B (NF-κB), heme oxygenase 1 (HMOX1), and glyceraldehyde 3-phosphate dehydrogenase (GAPDH). Primers were purchased from Macrogen, South Korea. The gene expression was normalized to the GAPDH gene and analyses were done using the comparative ΔΔCt method [49]. The mRNA level changes were expressed as fold change as compared to the sham animals.

## 5. Western blot analysis

Protein samples of cerebral cortices were extracted with RIPA lysis buffer (Pierce, Rockford, IL). Supernatants were collected and protein concentrations were measured using a Quick Start™ Bradford Protein Assay Kit from Bio-Rad. Protein samples (50 μg) were denatured for 10 min at 70°C, incubated on ice for 1 min, and electrophoresed on 10% sodium dodecyl sulfate-polyacrylamide gels for NLRP3 and SIRT1 blots and 12% for glial fibrillary acidic protein (GFAP), neuronal nuclei (NeuN), ionized calcium-binding adaptor molecule 1 (Iba1), Beta-tubulin (β-tubulin) and glyceraldehyde 3-phosphate dehydrogenase (GAPDH). Loaded

**Table 1. Primers sequence for the investigated genes.**

| Primer Name | Primer Sequence |
|---|---|
| NLRP3 Forward | 5′-GCTAAGAAGGACCAGCCAGAGT-3′ |
| NLRP3 Reverse | 5′-GAACCTGCTTCTCACATGTCGT-3′ |
| ASC Forward | 5′-TGCTTAGAGACATGGGCTTAC-3′ |
| ASC Reverse | 5′-CTGTCCTTCAGTCAGCACACT-3′ |
| Caspase1 Forward | 5′-GACAAGGCACGGGACCTATGT-3′ |
| Capase1 Reverse | 5′-CAGTCAGTCCTGGAAATGTGC-3′ |
| IL-18 Forward | 5′-TGGTTCCATGCTTTCTGGACTCCT-3 |
| IL-18 Reverse | 5′-TTCCTGGGCCAAGAGGAAGTGATT-3 |
| IL-1β Forward | 5′-GCCCATCCTCTGTGACTCAT-3′ |
| IL-1β Reverse | 5′-AGGCCAC AGGTATTTTGTCG-3′ |
| Nrf2 Forward | 5′- CCTCGCTGGAAAAAGAAGTG-3′ |
| Nrf2 Reverse | 5′-GGAGAGGATGCTGCTGAAAG-3′ |
| NF-κB Forward | 5′-GAAATTCCTGATCCAGACAAAAAC-3′ |
| NF-κB Reverse | 5′-ATCACTTCAATGGCCTGTGTGTAG-3′ |
| HMOX-1 Forward | 5′-CCTTCCCGAACATCGACAGCC-3′ |
| HMOX-1 Reverse | 5′-GCAGCTCCTCAAACAGCTCAA-3′ |
| GAPDH Forward | 5′-AACGACCCCTTCATTGAC-3′ |
| GAPDH Reverse | 5′- TCCACGACATACTCAGCAC-3′ |

samples were transferred to nitrocellulose blotting membranes 0.2μm from Amersham Protran™ (GE Healthcare Life science, Germany). Membranes were blocked with 5% blotting-grade blocker solution diluted in Tris-buffered saline containing 0.1% Tween-20 (TBST) (Bio-Rad, USA) for 1 hour. After blocking, membranes were incubated overnight at 4˚C with the following primary antibody: anti- Iba1, anti- Neu N diluted 1:1000 (EnCor Biotechnology, USA), anti-GFAP (EnCor Biotechnology, USA) diluted 1:5000, and anti-NLRP3, anti-SIRT1, anti- β-tubulin, and anti- GAPDH diluted 1:1000 (Cell Signaling Technology, USA). Membranes were washed 3 times 10 min each with TBST and incubated with horseradish peroxidase-conjugated anti-mouse IgG or anti-rabbit IgG (1:10000, Bio-Rad, USA) for 1 hour at room temperature. After washing with TBST 3 times (10 min each), enhanced chemiluminescence detection reagent (Pierce™ ECL Western, Thermo scientific, USA) was added to membranes to detect the immunoreactive protein bands on ChemiDoc MP Imaging System (Bio-Rad, USA). Bands intensity was quantified using ImageJ software.

## 6. Cytokines levels

The levels of the pro-inflammatory cytokines IL-1β and IL18 were measured using commercially available kits according to the manufacturer's instructions (Elabscience, United States, catalog number: E-EL-M0037 for IL-1β and E-EL-M0730 for IL-18). In brief, standards and supernatants of homogenates were added to the 96-well plates coated with the specific murine monoclonal antibodies raised against IL-1β or IL-18. Plates were then incubated for 90 min at 37˚C followed by decantation of the unbound antibodies. Secondary biotin-labeled specific antibodies for IL-1β and IL18 were added and incubated for 60 min at 37˚C. Following this incubation, the plates were washed with the supplied washing buffer before applying the HRP conjugate for 30 min at 37˚C. Finally, the substrate reagent was added, and the plates were incubated for 15 min at 37˚C followed by the addition of the stop reagent. The absorbance of the developed color was measured at 450 nm. The concentration of inflammatory cytokine in the brain tissues was extrapolated from the standard curve, expressed as pg/ml.

## 7. Statistical analysis

The statistical analysis was performed using GraphPad Prism version 8.0 (GraphPad Software, Inc., San Diego, CA). For the weight analysis, a two-way analysis of variance (ANOVA) was used followed by a Tukey test with a 95% confidence interval. For interpretation of the immunoblot, ELISA, and gene analysis results, a one-way analysis of variance (ANOVA) was used, followed by a Tukey test with a 95% confidence interval. All data were expressed as mean ± SEM of n = 9 per group. A P value $<0.05$ (*) was considered to indicate a statistically significant difference.

## Results

### Neurological severity score

To confirm TBI before assessing the effect of saffron, neurological severity score (NSS) was assessed at 1-hour and 24-hours post the final injury. The NSS of all sham animals was 0. Mice were able to perform completely all tasks at the two-time points proving their normal neurological behavior. However, the TBI group recorded an NSS of 5 indicating the occurrence of mild trauma, and it dropped to 1 after 24-hours. TBI mice treated with the saffron extract recorded an initial NSS score of 3 and dropped to zero after 24hours indicating the restoration of normal neurological behavior after saffron extract treatment.

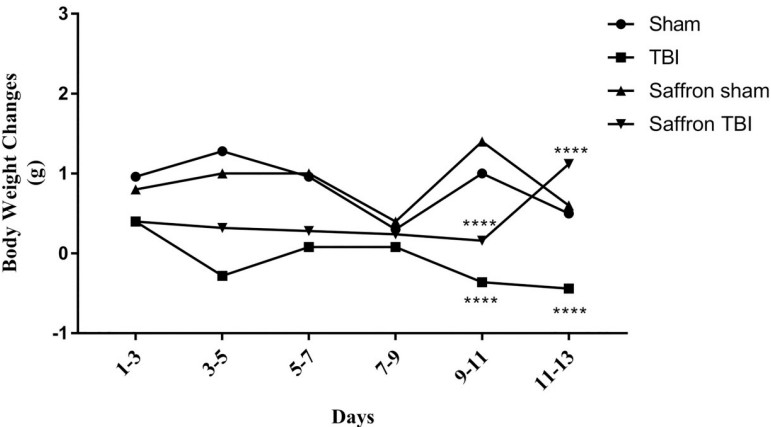

**Fig 2. The weight change between day 0 and day 13.** Sham, saffron sham, and saffron TBI mice were able to gain weight normally. TBI group failed to maintain normal body weights and start to lose weight significantly after Day 9. Data are mean ± SEM of 9 mice per group. (****) represent $P < 0.0001$.

## Effect of saffron on bodyweight

Weights of mice in all study groups (sham, saffron sham, TBI, saffron TBI) were recorded over the 13 days and the difference between day 1 and day 13 was analyzed (Fig 2). In the control groups, i.e., sham, and saffron sham, the average weights increased by 4g ($P > 0.5$) throughout the whole study. On the other hand, in the TBI group, the weight was maintained with no significant change until the 4th hit or after Day 7 ($P > 0.05$). However, after Day 9 or the 5th hit, significant average weight losses of 0.5g (F (3,96) = 24.62, $p < 0.0001$) were observed. Treating TBI mice with saffron extract induced significant weight gain of 1.12g (F (3,96) = 24.62, $p < 0.0001$) compared to the TBI group, implying its ability to prevent weight loss associated with the induced trauma. These results indicate that the brain injury pathological effects started after the 4th hit and it affected the baseline weights for mice and that saffron was powerful in ameliorating weight loss related to this injury.

## Saffron extract and repetitive injury didn't affect protein expression of the cytoskeletal proteins β-tubulin and GAPDH

Our main aim in our study is to investigate the involvement of NLRP3 in rmTBI. Safranal and crocin, the major saffron components [30] possess anti-tubulin activity [50,51]. TBI induces oxidative stress which may modify cytoskeletal proteins such as β-tubulin and β-actin [52]. Thus, to determine whether saffron or the injury-induced will modify the cytoskeletal protein, the levels of GAPDH and β-tubulin were quantified (Fig 3A). No significant changes were observed in GAPDH and β-tubulin levels in the four experimental groups implying that neither the brain injury nor the saffron treatment affected β-tubulin expression. Hence, both GAPDH and β-tubulin can be used as reference proteins.

## Saffron extract reduced protein expression levels of GFAP, NeuN, and Iba1 in the injured cortex following rmTBI

GFAP and Iba1 are major markers of astrocyte and microglial activity respectively [53,54] while NeuN acts as a neuronal damage marker [55]. Levels of these markers are used to assess the head injury. Any variations in the protein levels of these markers may reflect the astrocyte,

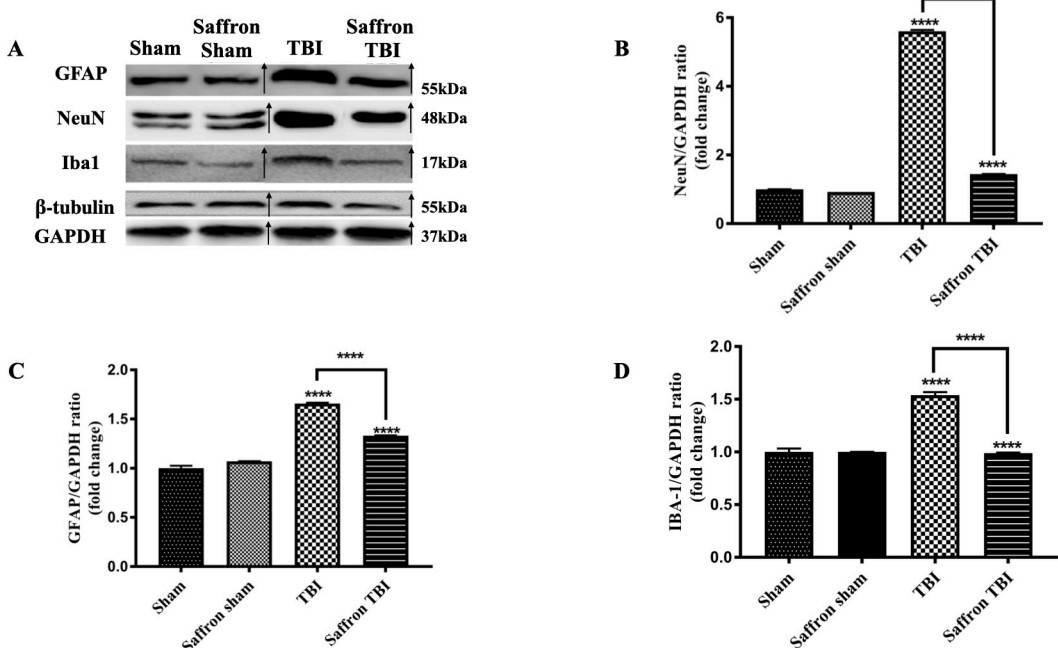

**Fig 3. Effect of saffron extract on GFAP, NeuN, Iba1 levels.** (A) is a composite representative western blot image of bands originating from different blots for GFAP, NeuN, Iba1, GAPDH, and β–tubulin. The bands of the TBI group and saffron TBI group were spliced from the same original image and re–ordered to be coherent as described in material and methods as samples were loaded differently. The arrow signs show the spliced and rearranged bands. Both β–tubulin and GAPDH were used as a loading control. (B), (C), and (D) show relative bar graphs of NeuN, GFAP, and Iba1 respectively relative to GAPDH 24hours after the final injury. Data are mean ± SEM of 9 mice per group. (****) represent $P < 0.0001$.

microglia, and neuronal state. A significant increase ($P < 0.0001$) in their levels was observed in the TBI group compared to normal as seen in Fig 3B–3D. Saffron treatment was able to significantly reduce these levels by 3 folds (F (3,32) = 11722, $p < 0.0001$), for NeuN, 0.4 folds (F (3,32) = 478.3, $p < 0.0001$) for GFAP and Iba1(F (3,32) = 156.3, $p < 0.0001$) compared to the untreated TBI group. As shown from these results, the induced brain injury may cause astrocyte and microglial activation, and saffron treatment mitigates these pathologies. Further immunohistological and morphological studies may be performed to study accurately these pathologies. The non-significant changes observed between the two sham groups emphasize the safety of the saffron dosage used.

## Saffron pretreatment reduced the mRNA expression and protein levels of NLRP3 in the injured cortex following rmTBI

After ensuring the activation of glial cells and inflammation occurrence, we determined the effect of rmTBI on the activation of NLRP3 inflammasome and evaluated the effect of the saffron extract on *NLRP3* gene expression and protein levels post TBI. As shown in Fig 4, no significant changes were detected in the saffron group. However, both the expression of *NLRP3* (Fig 4A) and its translated product (Fig 4B) significantly increased in the TBI group with 2.8 folds ($P < 0.0001$). Saffron treatment significantly ($P < 0.0001$) restored the levels of the inflammasome to normal both the transcriptional (F (3,32) = 13.11, $p < 0.0001$) and translation level (F (3,32) = 29493, $p < 0.0001$). These results suggest that NLRP3 inflammasome was activated after rmTBI and saffron extract significantly block the effect.

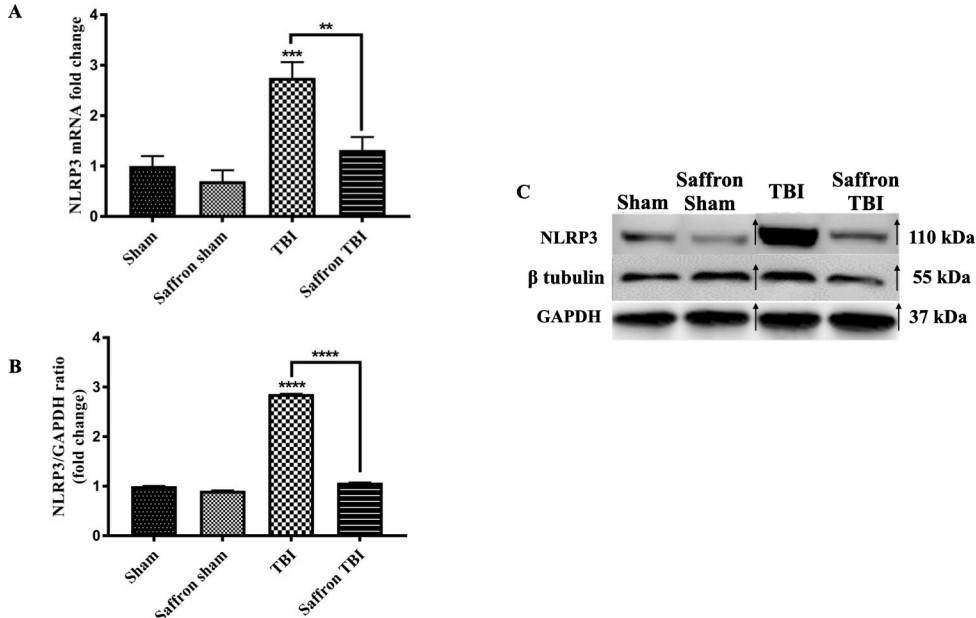

**Fig 4. Effect of saffron extract on NLRP3.** (A) and (B) show mRNA fold for NLRP3 and bar graphs of protein fold changes of NLRP3 to GAPDH respectively. (C) is a composite representative western blot image of bands originating from different blots of NLRP3, GAPDH, and β–tubulin. Bands of the TBI group and saffron TBI group were spliced from the same original image and re–ordered to be coherent as described in material and methods as samples were loaded differently, and with β–tubulin and GAPDH as loading controls 24hours post the final injury. The arrow signs show the spliced and rearranged bands. Data are mean ± SEM of 9 mice per group. (**), (***), (****) represent $P < 0.01$, $P < 0.001$, $P < 0.0001$.

## Saffron pretreatment reduced the mRNA expression of NLRP3 associated genes ASC, caspase 1, IL-1β, and IL-18 in the injured cortex following rmTBI

After assessing the NLRP3 gene and protein expressions, the levels of the associated genes *ASC, caspase 1* were measured along with their downstream effectors *IL-1β, and IL-18* (Fig 5). Saffron did not induce any change in the expression of these genes in normal mice compared to the sham group. However, their levels in TBI group increased significantly compared to the sham group where the expression of levels of *ASC* increased significantly by 5 folds (F (3,32) = 230.5, $p < 0.0001$) and 4 folds increase was observed in each, *caspase 1* (F (3,32) = 54.53, $p < 0.0001$), *IL-1β* (F(3,32) = 20.3, $p < 0.0001$), and *IL-18* (F(3,32) = 54.02, $p < 0.0001$). Treating TBI mice with saffron extract induced a significant decrease in expression by 3, 4, 1.3, and 1.9 folds ($P < 0.0001$) for *ASC, caspase 1, IL-1β, and IL-18* respectively.

Both IL-1β and IL-18, the effector molecules of NLRP3 inflammasome activation, are inflammatory initiating cytokines. Their protein levels were measured (Fig 5E and 5F) in all experimental groups. Saffron administration to normal mice (Saffron Sham) did not induce any change in their transcript levels. However, their levels in the cortical homogenates significantly increased in TBI group by 10 and 7 folds (F (3,20) = 20.3, $p < 0.0001$) and 7 folds (F (3,20) = 954.4, $p < 0.0001$) for IL-1β and IL-18 respectively. Treatment with saffron extract induced a notable decrease in the IL-1β and IL-18 cortex levels by 5 and 4.5 folds respectively ($P > 0.0001$) as compared to the untreated TBI group, correlated with their transcriptional levels.

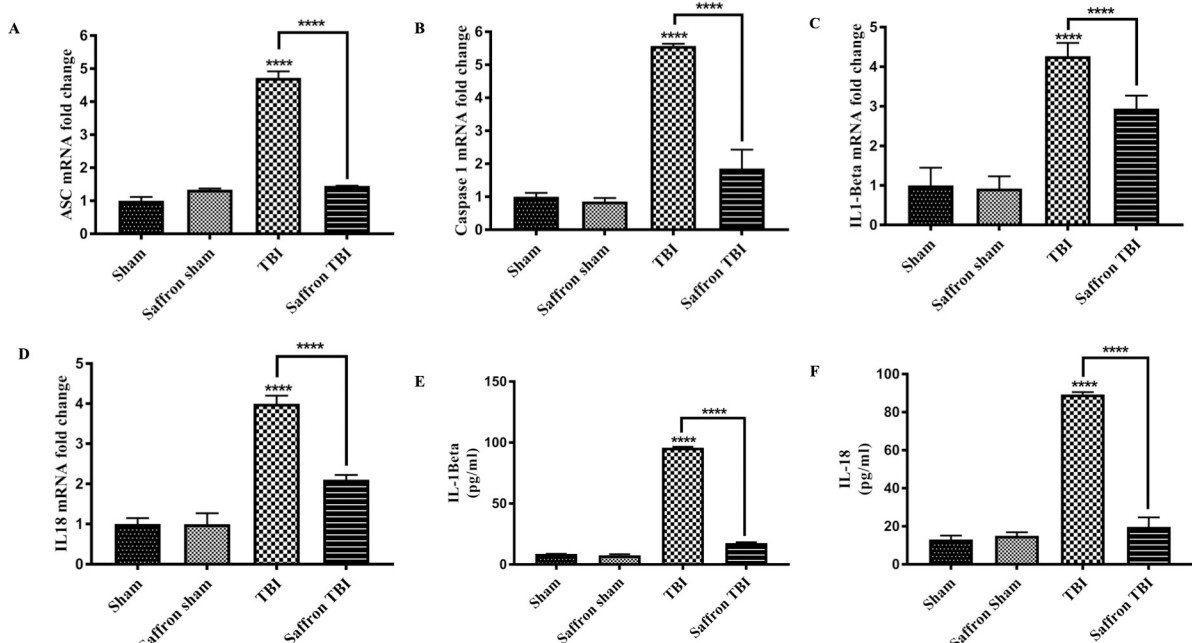

**Fig 5. Effect of saffron extract on ASC, caspase 1, IL-1β, and IL-18 levels.** (A), (B), (C), and (D) show mRNA fold–changes of ASC, caspase1, IL-1β, and IL-18 genes respectively. (E) and (F) show IL-1β and IL-18 levels in cortex homogenate respectively. Data are mean ± SEM of 9 mice per group. (****) represent $P < 0.0001$.

These results are consistent with the NLRP3 data obtained. Upregulation of *NLRP3* along with its associated genes *ASC* and *caspase 1* led to the increase in IL-1β and IL-18 both at the transcriptional and translation levels.

## Saffron extract attenuated NLRP3 inflammasome signaling activation via enhanced expression of SIRT1 in the injured cortex following rmTBI

To decipher the role of saffron in neuroinflammation attenuation through the NLRP3 inflammasome signaling, we examined the translational level of SIRT 1 and the transcriptional level of NF- κB. The NF-κB mRNA levels increased in the TBI group by 5.5 folds (F (3,32) = 120.5, $p < 0.0001$) 24-hours post the final injury when compared to the sham group (Fig 6A). Administration of saffron induced a significant 4 folds decrease (F (3,32) = 1998, $p < 0.0001$) in *NF-κB* levels.

At the translational level, SIRT1 levels significantly increased in the TBI group by 0.5 folds as part of its normal neuroprotective effect post-TBI. Saffron treatment was able to significantly increase further this level by 0.4 folds ($P < 0.0001$) (Fig 6B).

## Saffron extract upregulates mRNA expression of NRF2 and heme oxygenase-1 (HMOX1) in the injured cortex following rmTBI

Oxidative stress plays a key role in NLRP3 inflammasome activation during TBI. Reactive oxygen species (ROS) can stimulate NLRP3 inflammasome activation in the brain. To determine whether saffron extract and SIRT1 had an impact on oxidative stress in the injured cortex following rmTBI, we assessed the mRNA changes of NRF2 and its target gene HMOX1. Our results show that the levels of *NRF2* were remarkably elevated in the TBI group compared to the sham group by 2 folds (F (3,32) = 36.59, $p < 0.0021$) (Fig 7A) while no significant changes

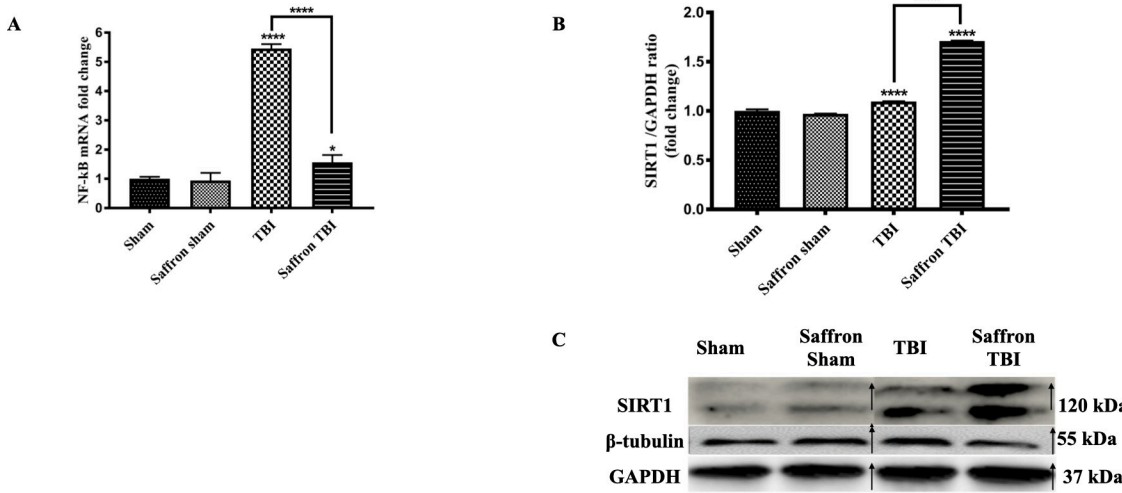

**Fig 6. Effect of saffron extract on NF–κB and SIRT 1.** (A) show the mRNA fold–change of NF–κB and (B) shows SIRT1 protein expression level relative to GAPDH. (C) is a composite representative western blot image of bands originating from different blots SIRT 1, GAPDH, and β–tubulin in which bands of the TBI group and saffron TBI group were spliced from the same original image and re–ordered to be coherent as described in material and methods as samples were loaded differently, and with β–tubulin and GAPDH as controls, 24 hours post the final injury. The arrow signs show the spliced and rearranged bands. Data are mean ± SEM of 9 mice per group. (*), (****) represent $P < 0.05$ and $P < 0.0001$.

were observed in *HMOX1* expression (Fig 7B). Saffron-treated TBI mice induced significant 4 folds increase in the levels of both genes (F (3,32) = 31.08, $p < 0.0001$) implying a protective role of saffron. Again, no significant differences between saffron sham and sham groups emphasizing the safety of the saffron dose used.

## Discussion

Neuroinflammation in different neurological abuses, including TBI, is a remarkable participant in dictating the illness evolution. Interest in studying mild variants of TBI is increasing due to the growing number of mTBI patients [56]. Although several models have been developed to explain rmTBI, there is no widely approved method to fully replicate human concussions. TBI anti-inflammatory therapeutic agents intend to dispose of dangerous signals, avoid the evolution of chronic neuroinflammation and ease the chronic regenerative immune

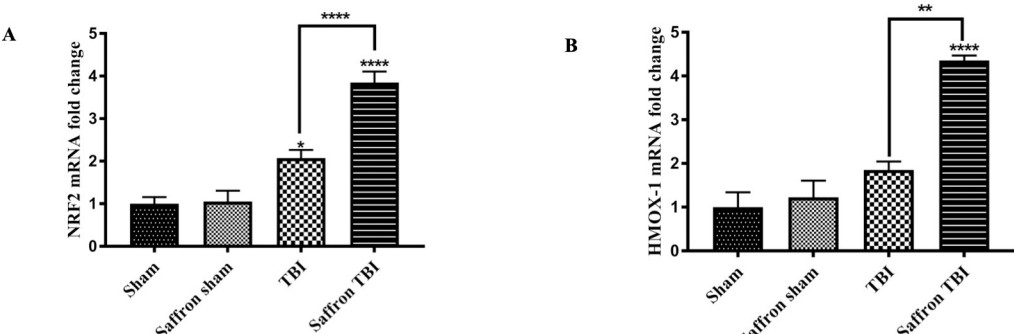

**Fig 7. Effect of saffron extract on NRF2 and HMOX1.** (A) and (B) shows the mRNA fold–change of NRF2 and HMOX1 respectively. Data are mean ± SEM of 9 mice per group. (*), (**) and (****) represent $P < 0.05$, $P < 0.01$ and $P < 0.0001$.

phenotype [57]. Even though many anti-inflammatory treatments have marked favorable effects in pre-clinical TBI models; however, transforming them into clinical effects had always been challenging. Although increased expression of NLRP3 inflammasome in the cortex can lead to an increase in the inflammatory response after TBI, further studies are essential to determine the exact function of NLRP3 inflammasome in TBI. Nevertheless, NLRP3 inflammasome can be considered a promising therapeutic target for TBI patients.

In this study, we investigated the role of the NLRP3 inflammasome and determined the neuroprotective effects of aqueous saffron extract in mitigating neuronal inflammation and oxidative stress following the rmTBI. To the best of our knowledge, no studies have examined the role of the NLRP3 inflammasome in the rmTBI closed head model although both pre-clinical and clinical studies have observed activation of NLRP1 and NLRP3 inflammasomes post-TBI [58,59]. In our experimental model, the non-invasive trauma induction process involved dropping a weight directly on an anesthetized mouse's head, which was left free to move after the impact. Following many trials of adjusting the height of the hit and inter-injury time, a successful TBI model was established. The injury protocol consisted of seven multiple hits for an inter-injury delay of 48 hours. We also reported that the change in the inter-injury time is associated with the severity of findings as seen by previous studies [60]. The mild feature of our repetitive injury model was manifested by the absence of any skull fracture and <5% mortality rate and confirmed by the NSS scores. Saffron treatment returned the mice to their normal behavior after 24 hours indicating its efficacy in restoring behavioral deficits associated with our injury model. Tracking weight changes in mice throughout the study showed that the injury affects mice weight and because saffron has proved to be a satiety enhancer [61], reduces blood biomarkers associated with obesity [62], and exert anti-obesity and body weight management effects in rats fed with high-fat diet [63]. Thus, we can assume that mice were able to gain weight due to the neuroprotective effect of saffron in restoring motor deficits which facilitated their access to food.

The glial cell, astrocyte, and microglial cells are the residential cells in CNS. They arose in response to TBI to exert their neuroprotective effects in improving TBI outcomes through their interactions. Reactive astrocytes can be classified into A1 and A2 phenotypes, which provide neuroprotective and neurotoxic effects, respectively. As part of its neuroprotective role, astrocytes, first can lessen glutamate excitotoxicity by reducing extracellular levels of glutamate [64], and enhance expression of neurotrophic factors such as brain-derived neurotrophic factor (BDNF) to reduce the injury-induced neuronal death, increase cell proliferation, and axonal repair [65]. Astrocytes can also regulate the ionic balance and cerebral blood flow [66], afford substrates needed in energy production for neurons [67], take separate in synapse development, and repair neuronal work [68]. Once TBI occurs, astrocytes will be among the primary cells that respond through astrogliosis [69]. The reactivity process is appended by cellular proliferation and hypertrophy along with increment levels in vimentin and GFAP [70]. The reactivity process also causes excessive production of matrix metalloprotease (MMP) that can affect BBB structure [71], and aquaporin 4 protein that is contributed for edema formation [72]. Glial scar formation is also contributed to astrocytes, in which after TBI, hypertrophic astrocytes are recruited to the site of damage and secrete chondroitin sulfate proteoglycans, an inhibitory cellular matrix [73]. The secreted matrix forms a physical and chemical barrier that can protect healthy brain tissue from the neurotoxins of the injury section, but this barrier still inhibits axonal repairment and growth [74]. Although astrogliosis is vital for axonal growth following the injury, the timeline of injury progression controls the advantages and disadvantages of astrogliosis to the CNS cells [75], in which prolonged astrocytes reactivity may hinder axonal regeneration and functional recovery [74]. GFAP, Iba-1, and Neu-N are different CNS injury biomarkers of astrocyte, microglia, neuronal cells

respectively [53–55]. The immunoblot results showed an increase in GFAP, Iba-1, and Neu-N folds in TBI mice confirming astrogliosis, microglial activation. Interestingly, the levels of NeuN increase post TBI, however, we didn't find any relevant studies that examine the immunoblot expression of this protein in a repetitive mild traumatic brain injury. Thus, further studies and analysis must be done to best describe this finding. The activated microglia passively and immediately release high mobility group box 1 (HMGB1) extracellularly HMGB1 that in turn binds toll-like receptor 4 (TLR4), initiating the MyD88-dependent pathway that leads to NF-κB activation, thus amplify further the signal for NLRP3 activation and intensifying the inflammatory response [76,77].

NLRP3 is an inflammasome complex comprised of NLRP3, ASC, and caspase 1 [59]. The principle aim of NLRP3 induction is to build a molecular platform for caspase 1 activation, leading later to the release of mature IL-1β and IL-18 and ultimate induction of the immune responses [78]. The molecular processes that govern NLRP3 inflammasome activation are yet a matter of controversy.

Neuroinflammation simulated in our model was assessed by NLRP3 inflammasome with its related genes *ASC, caspase 1, IL-1β*, and *IL-18*. Our findings showed that *NFκB* was upregulated post-TBI. The upregulated levels of *NF-κB* signal inflammasome priming [9]. The activated NF-κB induced increase in mRNA levels for NLRP3 along with *caspase 1, ASC*, and consequently *IL-1β*, and *IL-18*. Our data are coherent with the literature, for instance, two different studies following the weight drop TBI model defined the role of NLRP3 in TBI. The first study indicated that the cortex NLRP3 increased after 6 h for 7 days post-injury [79]. While the other study demonstrated a reduced inflammatory reaction and tissue destruction with retention of intellectual functions in the NLRP3 transgenic mouse model following TBI [80]. In a different TBI model, cortical impact injury model, the protein expression of NLRP3, caspase 1, and ASC raised in the early 7 days post-trauma, with peak expression after 3 days [81]. Collectively, these investigations besides our findings note that TBI promotes activation of inflammasomes, NLRP3 in particular.

Saffron treatment reduces the rise observed in NLRP3 at both the transcriptional and translational levels and a notable decline in the expression of its linked genes *caspase 1, ASC, IL-1β*, and *IL-1*. It is noteworthy to point out that the effect of the aqueous saffron extract had never been examined on NLRP3 expression neither in brain tissue nor other bodily organs. Similarly, this extract was never explored as well in the rmTBI mouse model and making it difficult to correlate our results with other investigators. However, we believe that the obtained results are due to the antioxidant and anti-inflammatory properties of saffron shown in other nervous system complications [82,83].

Studies on the neuroprotective effect of SIRT 1 in the TBI model and different neurological diseases are always in progress [17]. SIRT 1 is an endogenous neuroprotective factor and arbitrates protection via different pathways. It shows to regulate the p65 subunit of NF-κB and consequently regulates transcription of the downstream genes encoding different cytokines such as TNF-α, IL-8, and IL-6 [84]. Inhibition of SIRT1 deacetylase activity causes an increase in oxidative stress, inflammatory response [85], and neuronal apoptosis [86]. Besides its action on NF-κB, SIRT could affect neuronal inflammation by reducing astrocyte activation via suppressing the p38 mitogen-activated kinase (MAPK) signaling pathway in an experimental TBI model [87]. Our results showed that the expression of the astrocyte activation markers, GFAP, decreased following saffron administration which may be due to an increase in SIRT1 levels. In our study and a study done by Abedimanesh et al. saffron enhances SIRT1 expression [88]. Saffron instead of normalization after the injury, a potentiation of SIRT-1 levels in saffron-TBI Group was observed not in TBI brains. The Saffron sham group received saffron as well and both the saffron sham and sham group showed similar immunoblot results. In other words, we can say that saffron didn't increase SIRT1 expression until damage occurred in the tissue

after injury, and this increase is bestowed to the neuroprotective action exerted by saffron. This may open the door for investigations into the role of GFAP and SIRT1 via MAPK pathway in rmTBI for the treatment of post-traumatic inflammation.

NF-κB is not the only transcription factor that is regulated by SIRT 1. NRF2, the key player in the oxidative pathway, is under SIRT1 control too. SIRT1 upregulates the expression of NRF2 downstream genes such as those encoding for HMOX1, superoxide dismutase, and glutathione [89,90]. After repetitive mild brain injury, reactive oxygen species (ROS) deplete cellular antioxidants and interact with nucleotides and proteins, injured cells, and organelles such as neuronal cells, all together they intensify or aggravates inflammation [91]. For better analysis of the oxidative stress following injury, we investigated the mRNA levels of NRF2 and its downstream gene HMOX1. In the current study, the expression levels of NRF2 increased after TBI as a normal protective role. The results are similar to other studies that proved stimulation of the NRF2 pathway via oxidative stress after TBI [92]. Not only NRF2 but the expression of the HOMX1 gene is also upregulated in this study as observed by Salman et al. [93]. Saffron treatment enhanced the upregulation of NRF2, and HMOX1 to control the generated oxidative stress. The inhibition of the NRF2/HO1 pathways confirmed that our rmTBI model causes oxidative stress, saffron extract increased SIRT 1 expression which indirectly upregulates the NRF2/HO1 pathway.

In conclusion, we can state that our rmTBI model caused an increase in ROS production. The inflammatory response and oxidative stress were mediated by NF-κB and NRF2 activation. The inflammatory response was intensified in injured mice because of inflammasome activation, and an increase in oxidative stress. With saffron treatment, SIRT1 increased, leading to rising in mRNA levels of NRF2 and HMOX1, and a decrease in mRNA levels of NLRP3 inflammasome together with its related genes, IL-1β, and IL-18 (Fig 8). Drug discovery

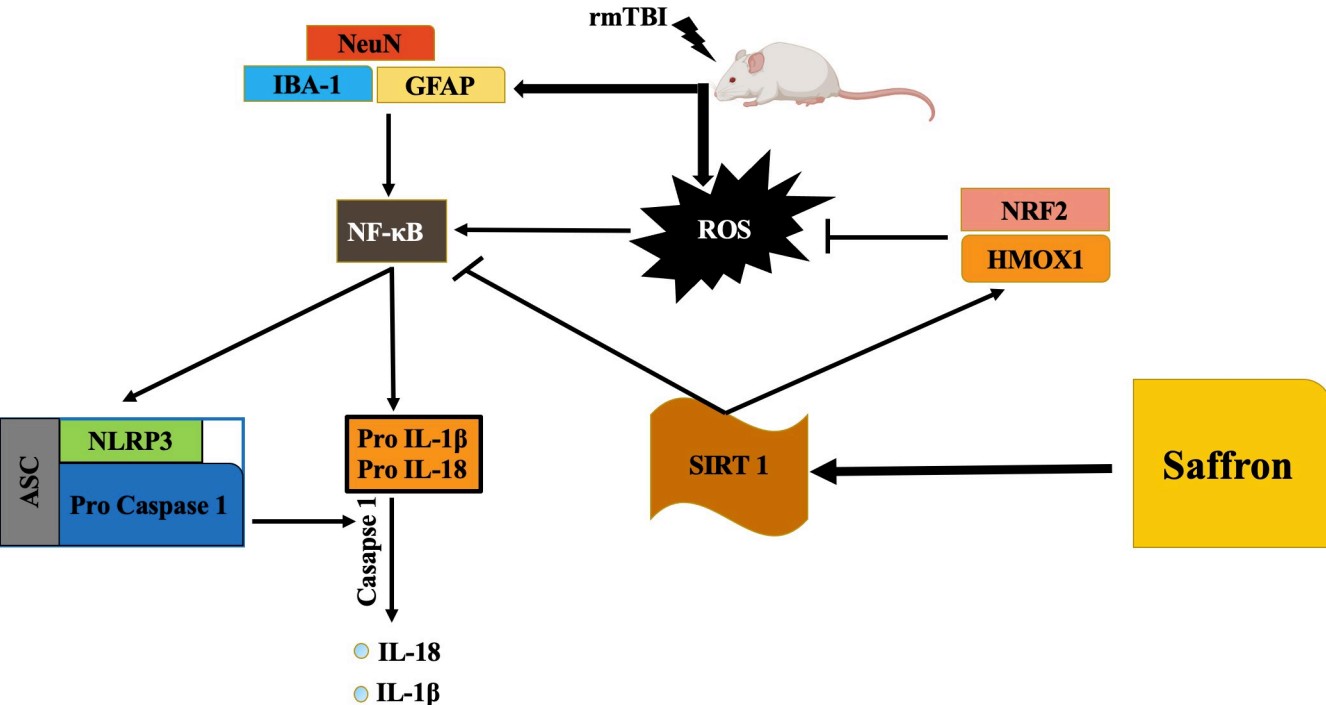

**Fig 8. Schematic illustration of saffron attenuating neuronal inflammation and oxidative stress post rmTBI, by suppressing NLRP3 inflammasome activation via SIRT1 activation** Abbreviations: ROS, reactive oxygen species; NLRP3, NOD–like receptors (NLR) family pyrin domain containing 3; ASC, apoptosis–associated speck–like protein containing a CARD; NF–κB, nuclear factor kappa B; IL–1β, interleukin–1Beta; IL–18, interleukin 18; SIRT1, sirtuin 1; NRF2, nuclear factor erythroid 2–related factor 2; HMOX1, heme oxygenase 1; GFAP, glial fibrillary acidic protein; Iba1, ionized calcium–binding adaptor molecule; Neu N, neuronal nuclei; rmTBI, repetitive mild traumatic brain injury.

targeting activation of NLRP3 inflammasome may be a promising therapeutic approach for TBI due to the critical function of NLRP3 inflammasomes in regulating neuroinflammatory reaction and neural tissue degradation following TBI.

## Supporting information

**S1 Raw images.**
(PDF)

## Author Contributions

**Conceptualization:** Mariam J. Shaheen, Jamilah M. Borjac.

**Data curation:** Mariam J. Shaheen.

**Formal analysis:** Mariam J. Shaheen.

**Funding acquisition:** Hana A. Itani.

**Methodology:** Amira M. Bekdash, Hana A. Itani.

**Supervision:** Jamilah M. Borjac.

**Validation:** Jamilah M. Borjac.

**Writing – original draft:** Mariam J. Shaheen.

**Writing – review & editing:** Hana A. Itani, Jamilah M. Borjac.

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
