## [Decision Letter · Decision Letter 0]

8 Jun 2021

PONE-D-21-13059

Saffron extract attenuates neuroinflammation in rmTBI mouse model by suppressing NLRP3 inflammasome activation via SIRT1

PLOS ONE

Dear Dr. Borjac,

Thank you for submitting your manuscript to PLOS ONE. After careful consideration, we feel that it has merit but does not fully meet PLOS ONE’s publication criteria as it currently stands. Therefore, we invite you to submit a revised version of the manuscript that addresses the points raised during the review process.

We look forward to receiving your revised manuscript.

Kind regards,

Faramarz Dehghani

Academic Editor

PLOS ONE

Journal Requirements:

2.Thank you for stating the following financial disclosure:

 "Disclosure statement uploaded."

3.PLOS ONE now requires that authors provide the original uncropped and unadjusted images underlying all blot or gel results reported in a submission’s figures or Supporting Information files. This policy and the journal’s other requirements for blot/gel reporting and figure preparation are described in detail at https://journals.plos.org/plosone/s/figures#loc-blot-and-gel-reporting-requirements and https://journals.plos.org/plosone/s/figures#loc-preparing-figures-from-image-files. When you submit your revised manuscript, please ensure that your figures adhere fully to these guidelines and provide the original underlying images for all blot or gel data reported in your submission. See the following link for instructions on providing the original image data: https://journals.plos.org/plosone/s/figures#loc-original-images-for-blots-and-gels.

Additional Editor Comments:

Dear Dr. Borjac,

I have now received the comments of the reviewers and agree with their points of criticism. Please conduct the suggested experiments and revise your manuscript accordingly. Furthermore, the presentation of gels in the supporting information (raw images) must be improved. Please add the molecular weights of markers and the labelling for the investigated groups in a unique way. Please recheck again the labelling of all protein bands as there are some mistakes. As an example in NLRP3 original gel saffron sham and sham do not match to the labelling in figure 4.

Reviewers' comments:

Reviewer's Responses to Questions

**Comments to the Author**

1. Is the manuscript technically sound, and do the data support the conclusions?

Reviewer #1: Yes

Reviewer #2: Yes

2. Has the statistical analysis been performed appropriately and rigorously? 

Reviewer #1: No

Reviewer #2: Yes

3. Have the authors made all data underlying the findings in their manuscript fully available?

Reviewer #1: Yes

Reviewer #2: No

4. Is the manuscript presented in an intelligible fashion and written in standard English?

Reviewer #1: Yes

Reviewer #2: Yes

5. Review Comments to the Author

Reviewer #1: The current paper presents data supporting the hypothesis that saffron, with its complex chemical composition, may cope with neurodegeneration induced by TBI. The initial event is always multifactorial but oxidative stress plays a pivotal role at various levels including the activation of neuroinflammatory pathways. Saffron acts at different levels directly as an antioxidant but also by regulating many genes and protein synthesis.

-Although the topic addressed in this submission is of potential interest, my initial evaluation is that the manuscript is not ready for consideration. Among the overarching concerns, the statistical analysis lacks sufficient detail, testing whether interaction effects between groups for the various outcome measures are significant. The results for the interaction terms in the overall ANOVAs are important to confirm that the effects on the various outcome measures of interest are in fact saffron-dependent, as the report implies.

- the most biologically active components are two carotenoids including crocin and crocetin. Most of the pharmacokinetic studies are related to these compounds. An open problem is to understand how saffron metabolites reach the CNS after i.p. application, is this a passive transcellular diffusion or an active one?

- It is not clear from the materials and methods which dose/volume of Saffron-Extract was i.p injected. It is mentioned on page 4 …the dose of the saffron extract used in this study was based on previous studies 32, 33???

- Neuronal severity score: why the data are not shown?

- The data concerning effects of saffron on body weight are amazing. in Hits 1 and 2, the differences between Sham/saffron sham and TBI are not significant, however they are significant in Hits 5, 6 and 7. How to explain that TBI group failed to maintain normal body weights and Saffron-TBI mice were able to gain weight. Is this due to the injury or to saffron. One can be assumed that injured mice have motoric difficulties to food-access.

Fig. 2: Hit1 TBI- 35 vs TBI+Saffron 32 this difference is not significant- Hit5: TBI approx. 32- TBI+Saffron 31 this difference is significant (p< 0.0001). Is it quite difficult to understand this statistic interpretation? The same remark for Hits 6 & 7.

- the weight differences are in the gram range; it cannot be ruled out that the edema could play a role. While Brain edema is a severe complication of TBI. The accumulation of intracellular water, especially in the perivascular astrocytes, disrupts the local osmotic environment, resulting in or aggravating the breakdown of the BBB.

- Fig. 3: Authors stated that saffron extract reduces neuronal damage, astrocyte’s proliferation and microglial activation which are TBI-mediated. This analysis has been done only on proteins-levels by WB. It should be emphasized that the increase in GFAP- NeuN- and Iba1-Levels does not mean automatically glia-proliferation, neuronal-protection and microglia-activation! These changes may indicate only alterations in protein-levels and not inevitably changes in the density of astrocytes, neurons and microglial-cells. Morphological data are needed in order to show that saffron reduced the TBI-mediated astrogliosis, neuronal damage and microglia activation. Furthermore, recent studies investigating neurodegenerative diseases reveals that neuroinflammation in the CNS induces the activation of two different paradigm of reactive astrocytes, termed A1 and A2. Notably, astrocyte reactivity is disease- and stimulus-dependent, adopting either a cytotoxic A1 phenotype or a neurotrophic, anti-inflammatory A2 phenotype. This point might be discussed.

How to explain the significant increase in NeuN-levels in the TBI-group comparing to sham-group on Fig. 3? If TBI induced neuronal injury, we should expect a decrease in neuronal density and consequently a reduction of NeuN-protein-levels!

-From Fig. 6: SIRT1- was significantly increased in TBI comparing to sham, but instead of normalization, we see a potentiation of SIRT-1 levels in saffron-TBI-Group. This is an interesting finding which needs a confirmation by using pharmacological inhibitors of saffron-uptake.

The same evolution concerns NRF2 and Hmox1 on mRNA-levels on Fig. 7. Here, why the data show only mRNA- and not protein-levels.

Reviewer #2: The manuscript, was found to be interesting on the exploring the anti-neuro-inflammatory effect of aquas (polar) extract

if the authors would have explained on mode of action on the base of compound level in the saffron extract , it will be more interesting and information on compound level

For example , at least partial purification based such as either it is a protein or peptide or either it is a polyphenol based compound it will be much more informative on the effect of the extract.

In efficacy and durability of the anti-neuroinflamatory property depends on the nature of the compound present in the saffron extract.

6. PLOS authors have the option to publish the peer review history of their article (what does this mean?). If published, this will include your full peer review and any attached files.

Reviewer #1: No

Reviewer #2: No

---

## [Author Response · Author response to Decision Letter 0]

1 Jul 2021

RESPONSE TO REVIEWERS

of “Saffron Extract Attenuates Neuroinflammation in rmTBI Mouse Model by Suppressing NLRP3 Inflammasome Activation via SIRT1”

Manuscript ID: PONE-D-21-13059

by Mariam Shaheen, Amira Bekdash, Hana A. Itani, Jamilah Borjac

We would like to thank the reviewers for their thorough review of the manuscript as they raise important issues and their comments are very helpful for improving the manuscript. We agree with almost all their comments and we have revised our manuscript accordingly. We included all reviewers’ suggestions and clarified the text when needed. We are confident that the new version of the manuscript is greatly improved. We respond below in detail to each of the reviewer’s comments. 

In addition, we hope that the reviewers find our responses to their comments satisfactory. Please, find below the reviewers comments repeated in italics and our responses inserted after each comment in italic blue. 

Looking forward to hearing from you soon. 

Sincerely, 

Authors

Response to comments

1. Among the overarching concerns, the statistical analysis lacks sufficient detail, testing whether interaction effects between groups for the various outcome measures are significant. The results for the interaction terms in the overall ANOVAs are important to confirm that the effects on the various outcome measures of interest are in fact saffron-dependent, as the report implies.

Reply: The authors did the needed corrections and added the related statistical data (Interaction F test) in the Result section in the following paragraphs/lines:

• Effect of Saffron on Bodyweight/ Lines 252-254

• Saffron extract reduces protein expression levels of GFAP, NeuN and Iba1 in the injured cortex following rmTBI /Lines 283-284

• Saffron pretreatment reduced the mRNA expression and protein levels of NLRP3 in the injured cortex following rmTBI/ Lines 308

• Saffron pretreatment reduced the mRNA expression of NLRP3 associated genes ASC, caspase 1, IL‐1β, and IL‐18 in the injured cortex following rmTBI /Lines 326-328, 335

• Saffron extract attenuated NLRP3 inflammasome signaling activation via enhanced expression of SIRT1 in the injured cortex following rmTBI/ Lines 353, 355

• Saffron extract upregulates mRNA expression of NRF2 and heme oxygenase-1 (HMOX1) in the injured cortex following rmTBI/ Lines 377,379

2. An open problem is to understand how saffron metabolites reach the CNS after i.p. application, is this a passive transcellular diffusion or an active one?

Reply: Saffron’s two major carotenoids, crocin and crocetin are the most biologically active elements. The primary site for carotenoids metabolism is the intestinal tract (1). In the intestine, crocin in the saffron is deglycosylated to trans-crocetin (2) enzymatically by the fecal microbiota (3). Trans-crocetin is then absorbed into the bloodstream via passive transcellular diffusion (3). Crocetin, in the bloodstream, is bound to albumin. The albumin-bound crocetin pushes through the blood brain barrier slowly enough to enter the CNS (3).

The authors added the above information in the Introduction section lines (103-108)

References:

1. Parker RS. 1996. “Absorption, Metabolism, and Transport of Carotenoids.” FASEB J. 10(5):542–51. 

2. Lautenschläger, M., J. Sendker, S. Hüwel, H. J. Galla, S. Brandt, M. Düfer, K. Riehemann, and A. Hensel. 2015. “Intestinal Formation of Trans-Crocetin from Saffron Extract (Crocus Sativus L.) and in Vitro Permeation through Intestinal and Blood Brain Barrier.” Phytomedicine 22(1). doi: 10.1016/j.phymed.2014.10.009. 

3. Asai, Akira, Takahisa Nakano, Masahiro Takahashi, and Akihiko Nagao. 2005. “Orally Administered Crocetin and Crocins Are Absorbed into Blood Plasma as Crocetin and Its Glucuronide Conjugates in Mice.” Journal of Agricultural and Food Chemistry 53(18). doi: 10.1021/jf0509355

3. It is not clear from the materials and methods which dose/volume of Saffron-Extract was i.p injected. It is mentioned on page 4 …the dose of the saffron extract used in this study was based on previous studies 32, 33???

Reply: The dose of saffron extract used was 50mg/kg. Animal studies on saffron, showed that a dose of 50mg/kg of aqueous saffron extract was safe and effective (4), and considered within the range of safe doses used in neurological related disorders and other diseases (4-9). 

The authors added the dose used with all the related in references in the manuscript, Material and Method section, paragraph 1 under “Preparation of saffron water extract” lines (140-142).

Reference:

4. José Bagur, M. et al. Saffron: An Old Medicinal Plant and a Potential Novel Functional Food. Molecules 23, (2017).

5. Amin, B., Moghri Feriz, H., Timcheh Hariri, A., Tayyebi Meybodi, N. & Hosseinzadeh, H. Protective effects of the aqueous extract of Crocus sativus against ethylene glycol induced nephrolithiasis in rats. EXCLI Journal 14, 411–422 (2015).

6. Hosseinzadeh, H., Abootorabi, A. & Sadeghnia, H. R. Protective Effect of Crocus sativus Stigma Extract and Crocin ( trans-crocin 4) on Methyl Methanesulfonate–Induced DNA Damage in Mice Organs. DNA and Cell Biology 27, 657–721 (2008).

7. Ettehadi, H. et al. Aqueous Extract of Saffron (Crocus sativus) Increases Brain Dopamine and Glutamate Concentrations in Rats. Journal of Behavioral and Brain Science 03, (2013).

8. Bostan, H. B., Mehri, S. & Hosseinzadeh, H. Toxicology effects of saffron and its constituents: A review. Iranian Journal of Basic Medical Sciences 20, 110–121 (2017).

9. Ziaee, T., Razavi, B. M. & Hosseinzadeh, H. Saffron Reduced Toxic Effects of its Constituent, Safranal, in Acute and Subacute Toxicities in Rats. Jundishapur Journal of Natural Pharmaceutical Products 9, 3–8 (2013).

4. Neuronal severity score: why the data are not shown? 

Reply: The results of the neuronal severity score (NSS) along with other behavioral tests are part of another manuscript submitted to Scientific Reports and is currently under revision. Calculation of the NSS was based on Zhuang et al. (10). The NSS was assessed at 1hour and 24hours post the final injury. The difference between the NSS at two different time points is a parameter that reflects injury severity post-TBI, as described by Chen et al. (11). The scores were classified as follows: 13-18 severe injury, 7-12 moderate injury,1-6 mild injury.

References:

10. Zhuang, Jian, Jie Li, Evelyn Ooi, Jonathan Bloom, Carrie Poon, Lax Daniel, Rosenbaum Daniel M, and Barone Frank C. 2015. “Modified Neurological Severity Score (MNSS) Tests and Scoring Values.” PLOS ONE.Dataset.

11. Chen, Yun, Shlomo Constsntini, Victoria Trembovler, Marta Weinstock, and Esther Shoham. 1996. “An Experimental Model of Closed Head Injury in Mice: Pathophysiology, Histopathology, and Cognitive Deficits.” Journal of Neurotrauma 13(10). doi: 10.1089/neu.1996.13.557.

5. In Hits 1 and 2, the differences between Sham/saffron sham and TBI are not significant, however they are significant in Hits 5, 6 and 7. How to explain that TBI group failed to maintain normal body weights and Saffron-TBI mice were able to gain weight. Is this due to the injury or to saffron. One can be assumed that injured mice have motoric difficulties to food-access.

Reply: In a separate study under this project, we tested the motor behaviors of TBI and Saffron TBI groups using the NSS score and other motor tests such as rotarod test, and pole climb test. Our results showed that the TBI injury caused motor deficits and that saffron-treatment restored motor functions. Despite the fact that saffron has proved to be satiety enhancer (12), reduces blood biomarkers associated with obesity (13), and exert anti-obesity and body weight management effects in rats fed with high fat diet (14), treating TBI mice with saffron allowed weight-gain due to its neuroprotective effect that aided in restoring motor deficits and thus facilitating their access to food. The authors emphasized on this comment in the Discussion section, lines 412-418.

Reference:

 12. Gout, B., Bourges, C. & Paineau-Dubreuil, S. Satiereal, a Crocus sativus L extract, reduces snacking and increases satiety in a randomized placebo-controlled study of mildly overweight, healthy women. Nutrition Research 30, (2010).

13. Ramli, F. N., Abu Bakar Sajak, A., Abas, F., Mat Daud, Z. A. & Azlan, A. Effect of Saffron Extract and Crocin in Serum Metabolites of Induced Obesity Rats. BioMed Research International 2020, (2020).

14. Mashmoul, M. et al. Effects of saffron extract and crocin on anthropometrical, nutritional and lipid profile parameters of rats fed a high fat diet. Journal of Functional Foods 8, (2014).

6. In Hit1 TBI- 35 vs TBI+Saffron 32 this difference is not significant- Hit5: TBI approx. 32- TBI+Saffron 31 this difference is significant (p< 0.0001). Is it quite difficult to understand this statistic interpretation? The same remark for Hits 6 & 7.

Reply: The determination of the p value depends on the average and the standard deviation of the variable tested. Therefore, based on the weigh averages and their SD of all groups, data analysis showed significant difference only in hit 5, 6 and 7. 

7. The weight differences are in the gram range; it cannot be ruled out that the edema could play a role. While Brain edema is a severe complication of TBI. The accumulation of intracellular water, especially in the perivascular astrocytes, disrupts the local osmotic environment, resulting in or aggravating the breakdown of the BBB.

Reply: Yes of course, brain edema is among the well noted features post brain injury. Unfortunately, we didn’t measure the total brain weight directly after dissection, but from our histological examination (results in the submitted manuscript to Scientific Reports under revision) edema was not noted which means the weight difference observed is not due to edema.

8. Authors stated that saffron extract reduces neuronal damage, astrocyte’s proliferation and microglial activation which are TBI-mediated. This analysis has been done only on proteins-levels by WB. It should be emphasized that the increase in GFAP- NeuN- and Iba1-Levels does not mean automatically glia-proliferation, neuronal-protection and microglia-activation! These changes may indicate only alterations in protein-levels and not inevitably changes in the density of astrocytes, neurons and microglial-cells. Morphological data are needed in order to show that saffron reduced the TBI-mediated astrogliosis, neuronal damage and microglia activation. 

Reply: We totally agree with the reviewers that WB data reflects alteration in proteins-levels and more experiments are needed. However, one of the major limitations to any study is the budget allocated. In this study, the first author had to pay for the majority of the kits needed in the project where this study constitutes only one part. Currently, the Lebanese currency crisis made it extremely difficult to add any additional experiments to relate the western blot results to immunohistochemistry studies or morphological studies. Based on the reviewer comment, we changed, in the Result section, the tittle of the paragraph entitled with “Saffron extract reduced neuronal damage, astrocyte and microglial activation in the injured cortex following rmTBI” to “Saffron extract reduces GFAP, NeuN and Iba1 protein levels in the injured cortex following rmTBI”, and emphasized within the same paragraph, lines 279-281, and 286-287, that western blot results on GFAP, NeuN, and Iba1-Levels does not necessarily reflect glia-proliferation, neuronal-protection and microglia-activation.

9. Furthermore, recent studies investigating neurodegenerative diseases reveals that neuroinflammation in the CNS induces the activation of two different paradigm of reactive astrocytes, termed A1 and A2. Notably, astrocyte reactivity is disease- and stimulus-dependent, adopting either a cytotoxic A1 phenotype or a neurotrophic, anti-inflammatory A2 phenotype. This point might be discussed.

Reply: The answer to the above comments is added in the Discussion section lines (420-440) as follow:

“The glial cells, microglial cells and astrocytes are the resident cells in the CNS. They arise in response to TBI to exert their neuroprotective effects to improve TBI outcomes through their interactions. Reactive astrocytes can be classified into A1 and A2 phenotypes that provide neuroprotective and neurotoxic effects respectively. As part of its neuroprotective role, astrocytes, first can lessen glutamate excitotoxicity by reducing extracellular levels of glutamate (12), and enhance expression of neurotrophic factors such as brain-derived neurotrophic factor (BDNF) to reduce the injury-induced neuronal death, increase cell proliferation, and axonal repair (13). Astrocytes can also regulate the ionic balance and cerebral blood flow (14), provide substrates needed in energy production for neurons (15), play a role in synapse development, and repair neuronal work (16). Once TBI occurs, astrocytes are among the primary cells that respond through astrogliosis (17). The reaction is linked to cellular proliferation and hypertrophy along with increment levels in vimentin and GFAP (18). It also causes excessive production of matrix metalloprotease (MMP) that can affect BBB structure (19), and aquaporin 4 protein that induces edema (20). Glial scar formation is also contributed to astrocyte, where after TBI, hypertrophic astrocytes are recruited to the site of damage, secrete the proteoglycan chondroitin sulfate, an inhibitory cellular matrix (21). The secreted matrix forms a physical and chemical barrier that can protect healthy brain tissue from the neurotoxins of the injury section, but this barrier still inhibits axonal repairment and growth (22). Although astrogliosis is vital for axonal growth following the injury, the timeline of injury progression controls the advantages and disadvantages of astrogliosis to the CNS cells (23), where prolonged astrocytes reactivity may hinder axonal regeneration and functional recovery (22).

Reference:

12. Sofroniew M v. Astrocyte barriers to neurotoxic inflammation. Nature Reviews Neuroscience. 2015 May 20;16(5). 

13. Zou P, Liu X, Li G, Wang Y. Resveratrol pretreatment attenuates traumatic brain injury in rats by suppressing NLRP3 inflammasome activation via SIRT1. Molecular Medicine Reports. 2018;17(2):3212–7. 

14. Baldwin KT, Eroglu C. Molecular mechanisms of astrocyte-induced synaptogenesis. Current Opinion in Neurobiology. 2017 Aug;45. 

15. Burda JE, Bernstein AM, Sofroniew M v. Astrocyte roles in traumatic brain injury. Experimental Neurology. 2016 Jan;275. 

16. Luo H, Wu X-Q, Zhao M, Wang Q, Jiang G-P, Cai W-J, et al. Expression of vimentin and glial fibrillary acidic protein in central nervous system development of rats. Asian Pacific Journal of Tropical Medicine. 2017 Dec;10(12). 

17. Cabezas R, Ãvila M, Gonzalez J, El-BachÃ¡ RS, BÃ¡ez E, GarcÃ¬a-Segura LM, et al. Astrocytic modulation of blood brain barrier: perspectives on Parkinsons disease. Frontiers in Cellular Neuroscience. 2014 Aug 4;8. 

18. Saadoun S. Involvement of aquaporin-4 in astroglial cell migration and glial scar formation. Journal of Cell Science. 2005 Dec 15;118(24). 

19. Cafferty WBJ, Yang S, Duffy PJ, Li S, Strittmatter SM. Functional Axonal Regeneration through Astrocytic Scar Genetically Modified to Digest Chondroitin Sulfate Proteoglycans. The Journal of Neuroscience. 2007;27(9):2176–85. 

20. Sofroniew M v. Molecular dissection of reactive astrogliosis and glial scar formation. Trends in Neurosciences. 2009;32(12):638–47. 

21. Toy D, Namgung U. Role of Glial Cells in Axonal Regeneration. Experimental Neurobiology. 2013 Jun 30;22(2). 

22. Morganti-Kossmann MC, Semple BD, Hellewell SC, Bye N, Ziebell JM. The complexity of neuroinflammation consequent to traumatic brain injury: from research evidence to potential treatments. Acta Neuropathologica. 2019 May 7;137(5). 

10. How to explain the significant increase in NeuN-levels in the TBI-group comparing to sham-group on Fig. 3? If TBI induced neuronal injury, we should expect a decrease in neuronal density and consequently a reduction of NeuN-protein-levels!

Reply: In our work we assessed the NeuN protein level 24h post the final injury and as far as we know, no studies assessed the protein level at this time point. The increased NeuN levels was observed in both TBI and Saffron TBI group. This finding needs further analysis and experimentation. In our model which is repetitive injury with only 48h interval between each hit, the injured neurons may attempt to compensate through its membrane capability of resealing all over the period of injury. We completely agree that western blot per se is not enough to confirm neuronal injury, however we thought that this distinctive notable increase of NeuN in TBI group and Saffron TBI must be mentioned and that it may open the way for new insights on understanding NeuN roles.

11. SIRT1- was significantly increased in TBI comparing to sham, but instead of normalization, we see a potentiation of SIRT-1 levels in saffron-TBI-Group. This is an interesting finding which needs a confirmation by using pharmacological inhibitors of saffron-uptake.

Reply: The main aim of our current study was to assess NLRP3 levels post rmTBI and examine the potential effects of saffron extract and the possible role or involvement of SIRT1 in this pathway. We totally agree with the reviewer that it would be quite interesting to find that saffron enhances SIRT1 expression and instead of normalization, it potentiated SIRT-1 levels in saffron-TBI-Group. Sirtinol which is a pharmacological inhibitor of saffron uptake could be used in later experiments to prove this point as used by Yang et al who showed the effect of SIRT1 in TBI (23). However, in our study, the Saffron sham group received saffron in the absence or occurrence of injury and both the saffron sham and sham group showed similar immunoblot results. Thus, we can assume that saffron by itself does not induce the increase in SIRT1 expression unless tissue damage occurs, i.e. post the injury. Using the pharmacological inhibitor would clarify the interplay between saffron, TBI, and SIRT1.

Reference: 

23. Yang, H. et al. SIRT1 plays a neuroprotective role in traumatic brain injury in rats via inhibiting the p38 MAPK pathway. Acta Pharmacologica Sinica 38, 168–181 (2017).

12. The same evolution concerns NRF2 and Hmox1 on mRNA-levels on Fig. 7. Here, why the data show only mRNA- and not protein-levels.

Reply: Quantifying the protein levels of both NRF2 and Hmox would definitely add more values to our results. However, due to the Lebanese currency crisis and the limited funds available, protein quantification was not performed. 

Reviewer #2:

 The manuscript, was found to be interesting on the exploring the anti-neuro-inflammatory effect of aquas (polar) extract if the authors would have explained on mode of action on the base of compound level in the saffron extract, it will be more interesting and information on compound level. For example, at least partial purification based such as either it is a protein or peptide or either it is a polyphenol-based compound it will be much more informative on the effect of the extract. In efficacy and durability of the anti-neuroinflammatory property depends on the nature of the compound present in the saffron extract.

Reply: The authors have added in the introduction section, line “99-102” and lines” 110-112” the following paragraph describing saffron major components, and the anti-inflammatory properties Saffron and its main active constituents.

“Phytochemical studies on saffron showed four major metabolites, crocin (80%), safranal (70%), and a modest percentage of picrocrocin and crocetin. The chemical analysis also revealed over 200 distinct compounds found in saffron including flavonoids, amino acids, vitamins (riboflavin and thiamine), anthocyanin, proteins, α-carotene, β-carotene, starch, zeaxanthin, mineral matter, and gums (24).

“Saffron exhibits antioxidant properties (25) and anti-inflammatory activities (26) via its major metabolites crocin (27,28), crocetin (29), and safranal (30). 

However, we believe that attempting to determine the effect of isolated compounds would be the scope of a new study. 

References:

 24. Alavizadeh, S. H. & Hosseinzadeh, H. Bioactivity assessment and toxicity of crocin: A comprehensive review. Food and Chemical Toxicology 64, (2014).

 25. Yoshino, F., Yoshida, A., Umigai, N., Kubo, K. & Chang, M. Crocetin reduces the oxidative stress induced reactive oxygen species in the stroke prone spontaneously hypertensive rats ( SHRSPs ) brain. J Clin Biochem Nutr 49, 182–187 (2011).

26. Christodoulou, E., Kadoglou, N. P., Kostomitsopoulos, N. & Valsami, G. Saffron: A natural product with potential pharmaceutical applications. Journal of Pharmacy and Pharmacology 67, 1634–1649 (2015).

 27. Kim, J.-H. et al. Crocin Suppresses LPS-Stimulated Expression of Inducible Nitric Oxide Synthase by Upregulation of Heme Oxygenase-1 via Calcium/Calmodulin-Dependent Protein Kinase 4. Mediators of Inflammation 2014, (2014).

 28. Xiong, Y., Wang, J., Yu, H., Zhang, X. & Miao, C. Anti-asthma potential of crocin and its effect on MAPK signaling pathway in a murine model of allergic airway disease. Immunopharmacology and Immunotoxicology 37, (2015).

 29. Diao, M., Min, J., Guo, F. & Zhang, C.-L. Effects of salbutamol aerosol combined with magnesium sulfate on T-lymphocyte subgroup and Th1/Th2 cytokines of pediatric asthma. Experimental and Therapeutic Medicine 13, (2017).

30. Farokhnia, M. et al. Comparing the efficacy and safety of Crocus sativus L. with memantine in patients with moderate to severe Alzheimer’s disease: a double-blind randomized clinical trial. Human Psychopharmacology: Clinical and Experimental 29, (2014).

---

## [Decision Letter · Decision Letter 1]

21 Jul 2021

PONE-D-21-13059R1

Saffron extract attenuates neuroinflammation in rmTBI mouse model by suppressing NLRP3 inflammasome activation via SIRT1

PLOS ONE

Dear Dr. Borjac,

Thank you for submitting your manuscript to PLOS ONE. After careful consideration, we feel that it has merit but does not fully meet PLOS ONE’s publication criteria as it currently stands. Therefore, we invite you to submit a revised version of the manuscript that addresses the points raised during the review process.

ACADEMIC EDITOR: Please insert comments here and delete this placeholder text when finished. Be sure to:

Indicate which changes you require for acceptance versus which changes you recommendAddress any conflicts between the reviews so that it's clear which advice the authors should followProvide specific feedback from your evaluation of the manuscript

We look forward to receiving your revised manuscript.

Kind regards,

Faramarz Dehghani

Academic Editor

PLOS ONE

Journal Requirements:

Additional Editor Comments (if provided):

Dear Dr. Borjac,

As you may notice, one of the reviewers was no longer available for judging the revised version of your manuscript. Therefore, the paper has been sent out to reviewer 3 who has raised additional concerns. The detailed suggestions are included below. Both reviewers ask extensive language editing. Please address the points of criticisms mentioned by the reviewers. Your work will be considered for a re-evaluation.

With best regards,

Faramarz Dehghani

Reviewers' comments:

Reviewer's Responses to Questions

**Comments to the Author**

1. If the authors have adequately addressed your comments raised in a previous round of review and you feel that this manuscript is now acceptable for publication, you may indicate that here to bypass the “Comments to the Author” section, enter your conflict of interest statement in the “Confidential to Editor” section, and submit your "Accept" recommendation.

Reviewer #1: All comments have been addressed

Reviewer #3: (No Response)

2. Is the manuscript technically sound, and do the data support the conclusions?

Reviewer #1: Yes

Reviewer #3: No

3. Has the statistical analysis been performed appropriately and rigorously? 

Reviewer #1: Yes

Reviewer #3: No

4. Have the authors made all data underlying the findings in their manuscript fully available?

Reviewer #1: Yes

Reviewer #3: Yes

5. Is the manuscript presented in an intelligible fashion and written in standard English?

Reviewer #1: Yes

Reviewer #3: No

6. Review Comments to the Author

Reviewer #1: Given financial problems as a reason for not having done this or that experiment is not logical in the scientific world. It is always possible to do so in a collaborative framework. In any case, one should never hesitate to ask for help elsewhere from colleagues who have the expertise and the means. I am willing to do it for you, but it seems that the authors are eager to see their paper published. In any case, the authors have taken my criticism into consideration, which collectively improve the quality of the manuscript.

Reviewer #3: The manuscript “Saffron extract attenuates neuroinflammation in rmTBI mouse model by suppressing NLRP3 inflammasome activation via SIRT1” by Shaheen et al. describes a potential neuroprotective effect of saffron extract. The authors describe a mouse model of traumatic brain injury and use qPCR and Western blot densitometry to follow marker genes and proteins that are upregulated upon brain injury. The authors potentially find protective effects of saffron upon traumatic brain injury.

The manuscript needs considerable proofreading and editing, which should not be the responsibility of a reviewer. Under no circumstance a manuscript should be send for revision in such condition, please send the manuscript to a native speaker or at least use spellchecking before submission. I will anyways try to list typos and editing issues:

47: insert space between 1 and (

79: insert space between (PRRs) and (9).

85: linked to the sirtuins should be to the sirtuin protein…

92-93: please check the sentence, it doesn’t make sense to me

104: sentence should be crocin is enzymatically deglycosylated…

107: BBB is mentioned for the first time, please described what it is (I guess the blood brain barrier)

108: insert space (23) The…

120-130: copy paste issue with different font size 125: …and the Canadian…(smaller size) and which of the experiments were performed in Canada?

128: anesthetizing is italics

273: …used as reference genes should be protein

312: should be mRNA fold change…

326: the expression levels (delete of)

333: …did not induced should be did not induce

334: change in their levels should be change in transcript level

341: …translational level not levels

369: mRNA expression should be mRNA levels

375: Our results shows should be show

376: compared with should be compared to

379: a protective role of saffron (delete play and insertof)

398: use target instead of aim

405: delete “anesthetized mouses brain” and write mouse head, left free to move… (you did not sedate the mouse brain, neither did you drop the weight on the mouse brain)

408: delete significantly (this is only used when statistics were performed)

Despite the formalities, the authors need to address several critical points:

- a detailed description of the saffron extract injected. What exactly did the authors inject? The aqueous 2% solution, the freeze dried precipitate or the re-solubilized freeze dried precipitate? Please try to be more clear in the description. Most ingredients in saffron won’t be water soluble, please discuss this in order to allow speculation which chemical substance in saffron might be responsible for neuroprotection.

- In Figure 2, I cannot see any label of the Y-axes (I guess this should be gramme (g)). The graph is not very straight forward and should be presented as body weight increase (Y) per time on X-axes. You can indicate the time of hit 1 and so on. Like this it looks like the hits lead to change in bodyweight. This should be presented as curves not as bar graphs. I am also confused by the statistics, from the graph it looks to me that there is absolutely no difference e.g. Hit7 TBI vs Saffron TBI. You show a significant change anyways!?

- 268: You write about anti-tubulin activity of saffron components. Please explain what you mean! As a biochemist, activity is reserved to enzymes, saffron is not an enzyme. Also what is anti-tubulin activity, does it interfere with polymerization, depolymerization, interferes with modification,…

- In Figure 3 it is unclear which band was quantified in the NeuN blot, there is a clear double band in the Sham and Saffron Sham but not in TBI. Which band did you use for quantification and how do you know which one is NeuN? Why is it not present in the TBI sample?

- Similarly, in Figure 6C it is unclear which band is SIRT1 and which one was used to quantify. The original blots should be presented without cropping in a supplementary figure.

- How do you explain an upregulation of SIRT1 only in TBI brains but not in Saffron sham mice.

The effect seems not dependent on Saffron. Please discuss this further.

7. PLOS authors have the option to publish the peer review history of their article (what does this mean?). If published, this will include your full peer review and any attached files.

Reviewer #1: No

Reviewer #3: No

---

## [Author Response · Author response to Decision Letter 1]

14 Aug 2021

RESPONSE TO REVIEWERS 

of “Saffron Extract Attenuates Neuroinflammation in rmTBI Mouse Model by Suppressing NLRP3 Inflammasome Activation via SIRT1”

Manuscript ID: PONE-D-21-13059

by M. Shaheen, A. Bekdash, H. Itani, and J. Borjac

We would like to thank the reviewers for their thorough review of the manuscript as they raise important issues and their comments are very helpful for improving the manuscript. We agree with almost all their comments, and we have revised our manuscript accordingly. We crafted a revised version of the paper that states the hypothesis and the implications of our work more clearly than before. Moreover, we included all reviewers’ suggestions and clarified the text when needed. We are confident that the new version of the manuscript will be greatly improved. We respond below in detail to each of the reviewer’s comments. 

In addition, we hope that the reviewers find our responses to their comments satisfactory. Please, find below the reviewers' comments repeated in italics and our responses inserted after each comment in italic blue. 

Looking forward to hearing from you soon. 

Sincerely, 

Authors

 

Reviewer #1: Given financial problems as a reason for not having done this or that experiment is not logical in the scientific world. It is always possible to do so in a collaborative framework. In any case, one should never hesitate to ask for help elsewhere from colleagues who have the expertise and the means. I am willing to do it for you, but it seems that the authors are eager to see their paper published. In any case, the authors have taken my criticism into consideration, which collectively improve the quality of the manuscript.

Reply: We would like to thank reviewer # 1 and we totally agree about the collaborative work between researchers. Indeed this work is a collaboration between Beirut Arab University and the American University of Beirut. We are indeed eager to publish this work as it is one of the graduation requirements for the first author of this manuscript who is a Ph.D. student. 

Reviewer #3: 

The manuscript needs considerable proofreading and editing, which should not be the responsibility of a reviewer.

Reply: The authors made all the needed editing

A detailed description of the saffron extract injected. What exactly did the authors inject? The aqueous 2% solution, the freeze-dried precipitate, or the re-solubilized freeze-dried precipitate? Please try to be more clear in the description. Most ingredients in saffron won’t be water-soluble, please discuss this to allow speculation which chemical substance in saffron might be responsible for neuroprotection.

Reply: A 2% aqueous solution was prepared by soaking one gram of ground saffron stigmas in 50ml of distilled water for 3 hours at room temperature in the dark. The filtrate was then freeze-dried yielding 0.065 g of a yellow-orange precipitate The precipitate was suspended in saline to a final concentration of 8.6% and used at a dose of 50mg/kg of body weight. The saffron dose used was based on animal studies showing its safety and effectiveness in neurologically related disorders and other diseases [1–6]. The phytochemical studies on saffron showed four major metabolites, crocin (80%), safranal (70%), and a modest percentage of picrocrocin and crocetin. These primary components modulate and manipulate saffron pharmacological effects and nutritional properties [7]. Yes, as the reviewer stated most ingredients in saffron are not water-soluble, but crocin is water-soluble. Crocin (C44H70O28), is the saffron’s major water-soluble stable carotenoid [8]. Many studies investigated and proved the neuroprotective effect of crocin in the TBI model [9], Alzheimer’s disease [10], neurotoxicity [11], cerebral ischemia [12], and in many different neurological disorders [13–18].

Reference:

1. Amin B, Moghri Feriz H, Timcheh Hariri A, Tayyebi Meybodi N, Hosseinzadeh H. Protective effects of the aqueous extract of Crocus sativus against ethylene glycol induced nephrolithiasis in rats. EXCLI Journal. 2015;14: 411–422. doi:10.17179/excli2014-510

2. Hosseinzadeh H, Abootorabi A, Sadeghnia HR. Protective Effect of Crocus sativus Stigma Extract and Crocin ( trans-crocin 4) on Methyl Methanesulfonate–Induced DNA Damage in Mice Organs. DNA and Cell Biology. 2008;27: 657–721. doi:10.1089/dna.2008.0767

3. Ettehadi H, Mojabi SN, Ranjbaran M, Shams J, Sahraei H, Hedayati M, et al. Aqueous Extract of Saffron (Crocus sativus) Increases Brain Dopamine and Glutamate Concentrations in Rats. Journal of Behavioral and Brain Science. 2013;03. doi:10.4236/jbbs.2013.33031

4. Bostan HB, Mehri S, Hosseinzadeh H. Toxicology effects of saffron and its constituents: A review. Iranian Journal of Basic Medical Sciences. 2017;20: 110–121. doi:10.22038/ijbms.2017.8230

5. Ziaee T, Razavi BM, Hosseinzadeh H. Saffron Reduced Toxic Effects of its Constituent, Safranal, in Acute and Subacute Toxicities in Rats. Jundishapur Journal of Natural Pharmaceutical Products. 2013;9: 3–8. doi:10.17795/jjnpp-13168

6. José Bagur M, Alonso Salinas G, Jiménez-Monreal A, Chaouqi S, Llorens S, Martínez-Tomé M, et al. Saffron: An Old Medicinal Plant and a Potential Novel Functional Food. Molecules. 2017;23. doi:10.3390/molecules23010030

7. Pandita D. Saffron (Crocus sativus L.): phytochemistry, therapeutic significance and omics-based biology. Medicinal and Aromatic Plants. Elsevier; 2021. doi:10.1016/B978-0-12-819590-1.00014-8

8. Alavizadeh SH, Hosseinzadeh H. Bioactivity assessment and toxicity of crocin: A comprehensive review. Food and Chemical Toxicology. 2014;64. doi:10.1016/j.fct.2013.11.016

9. Wang K, Zhang L, Rao W, Su N, Hui H, Wang L, et al. Neuroprotective effects of crocin against traumatic brain injury in mice: Involvement of notch signaling pathway. Neuroscience Letters. 2015;591: 53–58. doi:10.1016/j.neulet.2015.02.016

10. Lin L, Liu G, Yang L. Crocin Improves Cognitive Behavior in Rats with Alzheimer’s Disease by Regulating Endoplasmic Reticulum Stress and Apoptosis. BioMed Research International. 2019;2019: 1–9. doi:10.1155/2019/9454913

11. Rao SV, Hemalatha P, Yetish S, Muralidhara M, Rajini PS. Prophylactic neuroprotective propensity of Crocin, a carotenoid against rotenone induced neurotoxicity in mice: behavioural and biochemical evidence. Metabolic Brain Disease. 2019;34. doi:10.1007/s11011-019-00451-y

12. Zheng Y-Q, Liu J-X, Wang J-N, Xu L. Effects of crocin on reperfusion-induced oxidative/nitrative injury to cerebral microvessels after global cerebral ischemia. Brain Research. 2007;1138. doi:10.1016/j.brainres.2006.12.064

13. Shaterzadeh-Yazdi H, Samarghandian S, Farkhondeh T. Effects of Crocins in the Management of Neurodegenerative Pathologies: A Review. Neurophysiology. 2018;50: 302–308. doi:10.1007/s11062-018-9752-0

14. Farkhondeh T, Samarghandian S, Yazdi HS, Samini F. The protective effects of crocin in the management of neurodegenerative diseases: a review. Am J Neurodegener Dis. 2018. Available: www.AJND.us

15. Zhang X, Zhang X, Dang Z, Su S, Li Z, Lu D. Cognitive Protective Mechanism of Crocin Pretreatment in Rat Submitted to Acute High-Altitude Hypoxia Exposure. BioMed Research International. 2020;2020. doi:10.1155/2020/3409679

16. Yuan Y, Shan X, Men W, Zhai H, Qiao X, Geng L, et al. The effect of crocin on memory, hippocampal acetylcholine level, and apoptosis in a rat model of cerebral ischemia. Biomedicine & Pharmacotherapy. 2020;130. doi:10.1016/j.biopha.2020.110543

17. Naghizadeh B, Mansouri MT, Ghorbanzadeh B, Farbood Y, Sarkaki A. Protective effects of oral crocin against intracerebroventricular streptozotocin-induced spatial memory deficit and oxidative stress in rats. Phytomedicine. 2013;20. doi:10.1016/j.phymed.2012.12.019

18. Asalgoo S, Jahromi GP, Hatef B, Sahraei H. The Effect of Saffron Aqueous Extract and Crocin on PTSD Rat Models: The Focus on Learning and Spatial Memory. SSRN Electronic Journal. 2018. doi:10.2139/ssrn.3480403

In Figure 2, I cannot see any label of the Y-axes (I guess this should be gram (g)). The graph is not very straightforward and should be presented as body weight increase (Y) per time on X-axes. You can indicate the time of hit 1 and so on. Like this, it looks like the hits lead to a change in body weight. This should be presented as curves, not as bar graphs. I am also confused by the statistics, from the graph it looks to me that there is absolutely no difference e.g. Hit7 TBI vs Saffron TBI. You show a significant change anyways!?

Reply: Figure 2 has been redrawn as suggested by the reviewer. The difference in body weight changes is now clearer showing the significant changes after day 9 which corresponds to the 5th hit.

You write about the anti-tubulin activity of saffron components. Please explain what you mean! As a biochemist, activity is reserved for enzymes, saffron is not an enzyme. Also what is anti-tubulin activity, does it interfere with polymerization, depolymerization interferes with modification

Reply: Yes, the activity is definitely reserved for enzymes, and saffron is not an enzyme as you are specifying. However, Saffron’s main constituent, i.e., safranal, exerts its effects on cytoplasmic components through macromolecular synthesis and cytotoxicity [19]. Saffron is classified as an antioxidant [20] and anti-inflammatory [21] agent with antiproliferative effects that regulates tubulin activity [22]. The saffron anti-tubulin activity includes interference in tubulin polymerization where safranal binds to tubulin dimer between the alpha and beta subunit through hydrophobic interactions and one possible hydrogen bond leading to tubulin structural changes [23].

Reference:

19. Nair SC. Saffron Chemoprevention in Biology and Medicine: A Review. Biotherapy. 1995;10: 257–264. 

20. Nassiri-Asl M, Hosseinzadeh H. The role of saffron and its main components on oxidative stress in neurological diseases: A review. Oxidative Stress and Dietary Antioxidants in Neurological Diseases. Elsevier; 2020. doi:10.1016/B978-0-12-817780-8.00023-2

21. Fernández-Albarral JA, Ramírez AI, de Hoz R, López-Villarín N, Salobrar-García E, López-Cuenca I, et al. Neuroprotective and Anti-Inflammatory Effects of a Hydrophilic Saffron Extract in a Model of Glaucoma. International Journal of Molecular Sciences. 2019;20. doi:10.3390/ijms20174110

22. Bie X, Chen Y, Zheng X, Dai H. The role of crocetin in protection following cerebral contusion and in the enhancement of angiogenesis in rats. Fitoterapia. 2011;82. doi:10.1016/j.fitote.2011.06.001

23. Naghshineh A, Dadras A, Ghalandari B, Riazi GH, Modaresi SMS, Afrasiabi A, et al. Safranal as a novel anti-tubulin binding agent with potential use in cancer therapy: An in vitro study. Chemico-Biological Interactions. 2015;238. doi:10.1016/j.cbi.2015.06.023

In Figure 3 it is unclear which band was quantified in the NeuN blot, there is a clear double band in the Sham and Saffron Sham but not in TBI. Which band did you use for quantification and how do you know which one is NeuN? Why is it not present in the TBI sample?

Reply: The two bands at 48 and 46kDa correspond to the two isoforms of the NeuN/FOX3 protein as described by the antibody manufacturing company EnCor Biotechnology Inc. We quantified the upper band at 48kDa. The overexpression in the TBI group of the 48KDa band hid the lower band.

Similarly, in Figure 6C it is unclear which band is SIRT1 and which one was used to quantify. The original blots should be presented without cropping in a supplementary figure.

Reply: The original blots were submitted to the journal without any cropping. The two bands represent the SIRT1. The two bands could be for isoforms of SIRT1. We quantified the upper band observed at 120KDa as per the Cell signaling technology datasheet where we purchased the antibody. 

How do you explain an upregulation of SIRT1 only in TBI brains but not in Saffron sham mice? The effect seems not dependent on Saffron. Please discuss this further.

Reply: SIRT 1 is an endogenous neuroprotective factor that confers protection via different pathways. Saffron was observed to enhance SIRT1 gene expression by Abedimanesh et al [24] in patients with coronary artery diseases and our study an increase in SIRT-1 was observed protein levels in TBI and a further increase in the saffron-TBI group rather than simple normalization after the injury. The Saffron sham and the sham groups showed similar levels of SIRT-1 implying that saffron does not increase SIRT1 expression. The increase was observed after inducing the injury and thus this increase is associated with the neuroprotective action exerted by saffron. 

Reference:

24. Abedimanesh N, Motlagh B, Abedimanesh S, Bathaie SZ, Separham A, Ostadrahimi A. Effects of crocin and saffron aqueous extract on gene expression of SIRT1, AMPK, LOX1,NF‐κB,and MCP‐1 in patients with coronary artery disease: A randomized placebo‐controlled clinical trial. Phytotherapy Research. 2020;34. doi:10.1002/ptr.6580

---

## [Decision Letter · Decision Letter 2]

26 Aug 2021

Saffron extract attenuates neuroinflammation in rmTBI mouse model by suppressing NLRP3 inflammasome activation via SIRT1

PONE-D-21-13059R2

Dear Dr. Borjac,

We’re pleased to inform you that your manuscript has been judged scientifically suitable for publication and will be formally accepted for publication once it meets all outstanding technical requirements.

Kind regards,

Faramarz Dehghani

Academic Editor

PLOS ONE

Additional Editor Comments (optional):

Reviewers' comments:

Reviewer's Responses to Questions

**Comments to the Author**

1. If the authors have adequately addressed your comments raised in a previous round of review and you feel that this manuscript is now acceptable for publication, you may indicate that here to bypass the “Comments to the Author” section, enter your conflict of interest statement in the “Confidential to Editor” section, and submit your "Accept" recommendation.

Reviewer #3: All comments have been addressed

2. Is the manuscript technically sound, and do the data support the conclusions?

Reviewer #3: Yes

3. Has the statistical analysis been performed appropriately and rigorously? 

Reviewer #3: I Don't Know

4. Have the authors made all data underlying the findings in their manuscript fully available?

Reviewer #3: Yes

5. Is the manuscript presented in an intelligible fashion and written in standard English?

Reviewer #3: Yes

6. Review Comments to the Author

Reviewer #3: All comments have been adressed and the manuscript has improved accordingly. The manuscript can be processed for publishing.

7. PLOS authors have the option to publish the peer review history of their article (what does this mean?). If published, this will include your full peer review and any attached files.

Reviewer #3: No

---

## [Editor Report · Acceptance letter]

31 Aug 2021

PONE-D-21-13059R2 

Saffron extract attenuates neuroinflammation in rmTBI mouse model by suppressing NLRP3 inflammasome activation via SIRT1 

Dear Dr. Borjac:

I'm pleased to inform you that your manuscript has been deemed suitable for publication in PLOS ONE. Congratulations! Your manuscript is now with our production department. 

Kind regards, 

on behalf of

Dr. Faramarz Dehghani 

Academic Editor

PLOS ONE